# `HOBIT`: Hardness Optimized Batch Sampling for InfoNCE Training

**Himanshu Dutta** [1]  **Lokesh Nagalapatti** [1]  **Yashoteja Prabhu** [1]

## Abstract

Contrastive training with InfoNCE loss and in-batch negatives is the standard approach for learning dual-encoder models. Its effectiveness, however, critically depends on the availability of hard negatives; in their absence, learning quickly saturates. Existing methods address this via explicit hard-negative mining, which is often costly or heuristic-driven. We introduce `HOBIT`, a principled mini-batch construction method that improves in-batch negative quality by reordering training examples at every epoch. `HOBIT` solves an optimization problem motivated by the InfoNCE objective to yield mini-batches such that each query in the batch is exposed to hard yet non-contradictory, informative negative examples. We show that the optimization objective is monotone and submodular which in turn leads us to a greedy algorithm that admits the standard $\mathcal{O}(1-1/e)$ approximation guarantee. Empirically, we show that `HOBIT` incurs negligible computational overhead while significantly outperforming state-of-the-art batching methods, and remains complementary to existing hard negative mining techniques.

## 1. Introduction

Dual encoders are a prevalent architecture for building large-scale information retrieval systems. They encode queries and documents independently into a shared embedding space, where relevance is estimated using a similarity measure such as cosine similarity or dot product. Such models are trained using contrastive learning to bring representations of relevant query–document pairs closer than irrelevant ones. The contrastive loss used for training depends on the supervision available in the data. When explicit labels for both positive and negative documents are provided for each query, one can directly optimize losses (Gupta et al., 2024b;

[1]Microsoft Research, India. Correspondence to: Himanshu Dutta <hdutta1024@gmail.com>.

*Proceedings of the 43rd International Conference on Machine Learning*, Seoul, South Korea. PMLR 306, 2026. Copyright 2026 by the author(s).

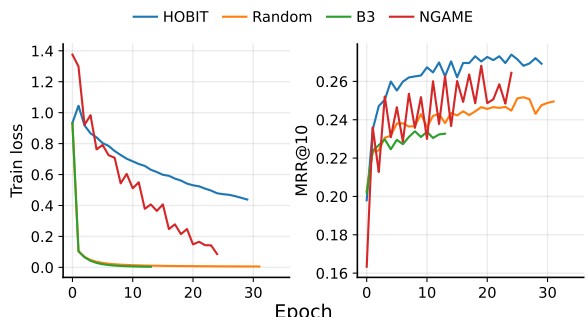

*Figure 1.* Training dynamics on MSMARCO illustrating the importance of hard negatives in InfoNCE training. Left: epoch-wise training loss. Right: validation MRR@10. Methods that rely on static or weakly adaptive batch construction (Random, B3) saturate early, while adaptive hard-negative batching methods (NGAME and `HOBIT`) sustain learning and achieve better generalization.

Robinson et al., 2021) that exploit this richer supervision. However, real-world retrieval datasets provide supervision only for positive query–document pairs, offering little to no guidance on what constitutes a meaningful negative.

In this weakly supervised regime, training typically relies on InfoNCE-style losses with in-batch negatives (Karpukhin et al., 2020; Qu et al., 2021; Gao & Callan, 2021; 2022), where positives of other queries within the same mini-batch are treated as negatives. While effective, this paradigm succeeds only when the in-batch negatives are sufficiently hard (Lin et al., 2021a; Zhan et al., 2021; Robinson et al., 2021); when batches are dominated by easy or redundant negatives, learning quickly saturates (see Fig. 1).

A large body of prior work has therefore focused on hard-negative mining. Most existing approaches either rely on expensive multi-stage training pipelines (Xiong et al., 2021; Sun et al., 2022) or select negatives using heuristic criteria (Dahiya et al., 2023; Yang et al., 2023). Although such methods yield performance gains, they significantly increase the training cost. A common design pattern is to maintain an approximate nearest-neighbor index over document embeddings and query it during training to retrieve hard negatives. However, because document representations evolve continuously as the model trains, such indices quickly become stale and require frequent rebuilding to remain aligned with the current model state (Xiong et al., 2021).

At scale, this indexing cost becomes prohibitive. For ex-

ample, MSMARCO (Nguyen et al., 2016) roughly contains only 500k query–positive pairs compared to 8 million corpus documents, making per-epoch document index reconstruction impractical. As a result, most methods rely on static or stale indices. Approaches such as ANCE (Xiong et al., 2021) partially mitigate this by refreshing the index every few epochs. However, a recent study (Monath et al., 2023) shows that stale negative mining biases gradient updates, effectively optimizing a surrogate objective that can significantly diverge from the true training objective. Consequently, balancing computational efficiency with training correctness remains a key challenge in hard-negative mining.

In this paper, we step back and ask a simpler question: *Can the batch itself be made informative?* We introduce **HOBIT**, a lightweight and principled batching strategy that deliberately constructs mini-batches so that examples serve as hard and informative negatives for one another. For MSMARCO, HOBIT optimizes over the 500k $(q, d)$ training pairs, making it orders of magnitude cheaper than approaches like ANCE, which must search over the entire 8 million–document corpus for each query in the batch.

By analyzing the gradient of the InfoNCE loss with respect to a query embedding $\vec{q}_i$, we show that an informative mini-batch should include negative documents $\vec{d}_j$ (paired with other queries in the batch) that satisfy two complementary properties relative to the query–document pair $(q_i, d_i)$. Specifically, such negatives should be ($i$) *hard*, exhibiting high query–document similarity ($\vec{q}_i^\top \vec{d}_j \uparrow$), while remaining ($ii$) *non-contradictory*, i.e., well separated from the positive document ($\vec{d}_i^\top \vec{d}_j \downarrow$). Exhaustive selection of examples that jointly satisfy these criteria across a mini-batch leads to a combinatorial optimization problem over a large discrete space, which is NP-hard. We show, however, that HOBIT induces a monotone submodular objective, enabling an efficient greedy optimization algorithm with a $\mathcal{O}(1 - 1/e)$ approximation guarantee. Our theoretical analysis further shows that this objective faithfully reflects the training dynamics of InfoNCE, providing a principled link between batch construction and contrastive learning.

Through extensive experiments, we show that HOBIT consistently outperforms state-of-the-art batching baselines across three datasets and two encoder architectures. On MSMARCO Passage (Nguyen et al., 2016), it improves MRR@10 by $+0.6$ points with roberta-base (Liu et al., 2019) and by $+1.5$ points with the stronger e5-large-unsupervised encoder (Wang et al., 2022). Gains are larger on Natural Questions (Kwiatkowski et al., 2019), where Recall@10 increases by $+3.3$ and $+1.6$ for the two encoders, respectively. Similar trends hold on TriviaQA (Joshi et al., 2017), with Recall@10 improvements of $+1.0$ and $+1.7$. Beyond outperforming stan-

dalone batching strategies, HOBIT complements explicit hard-negative mining, remains effective under sparse relevance supervision, and admits a cached variant, HOBIT-C, that achieves up to $29\%$ per-epoch training speedups with negligible impact on retrieval quality, making principled batch construction practical at scale.

**Contributions.** Our main contributions are:

1. We theoretically analyze InfoNCE training and formalize two complementary criteria for informative in-batch negatives: hardness and non-contradiction (Section 4; Appendix A.3).
2. We propose HOBIT, a principled batch sampling method that constructs informative mini-batches aligned with the InfoNCE objective (Sections 5–5.4).
3. We introduce HOBIT-C, an efficient cached approximation to HOBIT, and provide formal bounds on the approximation error induced by stale embeddings (Section 6.7; Appendix A.6).
4. We empirically show that HOBIT consistently outperforms state-of-the-art batch sampling methods across datasets and encoders, complements hard-negative mining, remains robust to sparse relevance supervision, and enables substantial training speedups via caching (Sections 6.3–6.7; Appendices A.8–A.13).

## 2. Motivating the Need for Hard Negatives

We motivate the need for having in-batch hard negatives using a simple experiment on MSMARCO. Figure 1 plots the training loss and validation performance (MRR@10) across epochs for four batch sampling methods: **Random**, which samples mini-batches uniformly at random; **B3**, which determines a fixed batch order once before training using a teacher model; **NGAME**, which adaptively reorders batches at each epoch based on clustering query embeddings; and **HOBIT**, our method. As shown in the left panel, training with Random and B3 quickly reaches a plateau. In contrast, NGAME and HOBIT, both of which adapt batch construction to the evolving encoder representations, maintain a consistently higher training loss over many epochs. This loss trend reflects the sustained presence of challenging in-batch negatives that continue to provide strong gradient signals to the model. This difference in training dynamics directly translates to the validation performance shown in the right panel. While Random and B3 yield poor validation MRR, adaptive methods, by contrast, achieve substantially better retrieval performance, with HOBIT consistently outperforming NGAME. While NGAME samples batches based on the simple heuristic criteria of query-query similarity, HOBIT optimizes a principled batching objective. These observations strongly motivate the need for a principled, training loss-aware mining of negative examples, which we develop in the following sections. We discuss a qualitative example

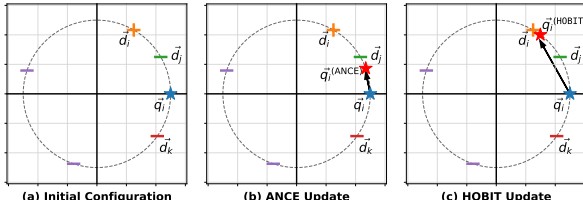

*Figure 2.* **Effect of negative selection on the query embedding.** All vectors are unit-normalized and lie on the unit circle. **(a)** Initial configuration showing a query $\vec{q}_i$, its positive document $\vec{d}_i$, a near-positive hard negative $\vec{d}_j$, and a distant hard negative $\vec{d}_k$. **(b)** Update induced by selecting the near-positive negative $\vec{d}_j$ using query–document similarity-based hard negative mining, as in ANCE (Xiong et al., 2021). $\vec{q}_i^{(\text{ANCE})}$ is the updated query. **(c)** Update induced by selecting the distant negative $\vec{d}_k$ as favored by HOBIT. Selecting negative based on Eq. 6 favours $\vec{d}_k$ over $\vec{d}_j$ for $\alpha \geq 0.5$. $\vec{q}_i^{(\text{HOBIT})}$ is the updated query. Updates are performed with the learning rate of 1.0.

in detail in Appendix A.1.

## 3. Related Work

We review prior work on mini-batching in this section, and defer a detailed discussion on dual-encoder training and hard-negative mining to Appendix A.2.

**Batching methods.** Batch construction has received far less attention in the literature compared to hard-negative mining, despite its ability to improve retrieval performance without full-corpus indexing. Most existing methods form batches by clustering queries or query–document interaction embeddings, thereby aiming to increase the hardness of in-batch negatives (Dahiya et al., 2023; Yang et al., 2023; Thirukovalluru et al., 2025). These studies demonstrate that batch composition alone can significantly enhance retrieval quality; however, the resulting batching objectives are largely heuristic-driven and not explicitly tied to the training objective. Complementary work exploits batch structure and interaction graphs to increase the diversity and semantic coherence of in-batch negatives (Lin et al., 2021b; Moiseev et al., 2023; Morris & Rush, 2025; Sachidananda et al., 2023; Thirukovalluru et al., 2025). Collectively, these works position principled batch construction as a lightweight and practical alternative to hard-negative mining.

## 4. Problem Formulation

Let $\mathcal{Q}$ denote the set of queries and $\mathcal{D}$ the document corpus from which relevant results are retrieved. Our training dataset $D_{\text{trn}} = \{(q_i, d_i)\}_{i=1}^N$ comprises $N$ query–positive pairs, where each training instance consists of a query $q_i \in \mathcal{Q}$ and an associated relevant document $d_i \in \mathcal{D}$. When a query has multiple relevant documents, it appears multiple times in $D_{\text{trn}}$, once for each positive.

Our goal is to learn a dual encoder model $f_\theta : \mathcal{Q} \cup \mathcal{D} \rightarrow \mathbb{R}^d$ that embeds queries and documents into a shared $d$-dimensional vector space. The training objective is to ensure that each query is more similar to its relevant documents than to any irrelevant document in the corpus. We assume that $f_\theta$ produces unit-normalized embeddings and assess relevance for a q-d pair using cosine similarity. Because $\mathcal{D}$ is large in realistic retrieval datasets, dual encoders are typically trained using the InfoNCE loss with in-batch negative sampling, which we describe next.

**InfoNCE Objective.** Given a mini-batch $B = \{(q_i, d_i)\}_{i=1}^b \subset D_{\text{trn}}$, in-batch contrastive training treats the positive documents associated with other queries in the same batch as negatives. Specifically, for each query $q_i$, its paired document $d_i$ is the positive example, while the documents $\{d_j\}_{j \neq i}$ in $B$ act as negatives. Let $\vec{q} = f_\theta(q)$ and $\vec{d} = f_\theta(d)$ denote the query and document embeddings, respectively. The similarity for a pair $(q_i, d_j)$ is defined as:

$$s_{ij} = f_\theta(q_i)^\top f_\theta(d_j) = \vec{q}_i^\top \vec{d}_j. \tag{1}$$

For a temperature parameter $\tau > 0$, per-query InfoNCE loss is

$$\mathcal{L}_i = -\log \frac{\exp(s_{ii}/\tau)}{\sum_{j=1}^b \exp(s_{ij}/\tau)} \tag{2}$$

Defining the softmax normalized similarity scores as

$$P_{ij} = \frac{\exp(s_{ij}/\tau)}{\sum_{k=1}^b \exp(s_{ik}/\tau)}, \tag{3}$$

the loss in Eq. 2 simplifies to $\mathcal{L}_i = -\log P_{ii}$.

To assess how batch composition influences learning, we begin our analysis by examining the gradient of the InfoNCE loss with respect to the query embedding.

**Lemma 4.1** (Gradient w.r.t. query embedding)**.** *For the InfoNCE loss $\mathcal{L}_i$, the gradient with respect to the query embedding $\vec{q}_i$ is given by $\nabla_{\vec{q}_i} \mathcal{L}_i = \frac{1}{\tau} \left( \sum_{j=1}^b P_{ij} \vec{d}_j - \vec{d}_i \right),$*

*Proof deferred to Appendix A.3.1.* □

*Remark* 4.2. Lemma 4.1 shows that learning is governed by competition between the positive document embedding $\vec{d}_i$ and a softmax-weighted average of batch documents. When a mini-batch contains only easy negatives, i.e., $s_{ii} \gg s_{ij}$ for all $j \neq i$, the softmax collapses to $P_{ii} \approx 1$ and $P_{ij} \approx 0$. As a result, the two terms in the gradient expression nearly cancel, yielding $\nabla_{\vec{q}_i} \mathcal{L}_i \approx 0$ and learning stalls.

**Theorem 4.3** (Dominant negative)**.** *Fix a query $q_i$ and suppose the in-batch softmax distribution satisfies*

$$P_{ii} = 1 - p - \epsilon, \qquad P_{ij} = p, \qquad \sum_{k \notin \{i,j\}} P_{ik} = \epsilon, \tag{4}$$

*for some $j \neq i$, with $p \in (0, 1)$ and $\epsilon \in (0, 1 - p)$. Then the gradient of the InfoNCE loss satisfies*

$$\left\| \nabla_{\vec{q_i}} \mathcal{L}_i \right\|_2 \; \geq \; \frac{p}{\tau} \sqrt{2(1 - t_{ij})} \; - \; \frac{2\epsilon}{\tau}. \tag{5}$$

*where, $t_{ij} = \vec{d_i}^{\top} \vec{d_j}$.*

*Proof deferred to Appendix A.3.2.* □

*Remark* 4.4 (Desiderata for an effective mini-batch). For a training pair $(q_j, d_j)$ in a mini-batch to provide a non-trivial learning signal for the loss incurred by $(q_i, d_i)$ under the InfoNCE objective, the corresponding in-batch negative $d_j$ must satisfy two complementary conditions:

1. *Hard:* it must attract a non-negligible softmax probability mass, i.e., the weight $p$ should be large. This requires the query-negative similarity $s_{ij} = \vec{q_i}^{\top} \vec{d_j}$ to be high.
2. *Non-contradictory:* it must be well separated from the positive document, meaning the similarity $t_{ij} = \vec{d_i}^{\top} \vec{d_j}$ should be small. Negatives that are near-duplicates of the positive are ambiguous and produce weak directional gradients, even when they are hard.

**Definition 4.5** (Hardness Score). We quantify the usefulness of an in-batch item $(q_j, d_j)$ for another item $(q_i, d_i)$ using the following measure:

$$w_{ij} \; = \; \vec{q_i}^{\top} \vec{d_j} - \alpha \vec{d_i}^{\top} \vec{d_j}, \tag{6}$$

where $\alpha > 0$ controls the trade-off between hardness and non-contradiction.

**Geometric Motivation for $w_{ij}$.** Figure 2 provides a geometric intuition for the pairwise hardness score $w_{ij}$. Panel (a) shows a query $\vec{q_i}$, its positive document $\vec{d_i}$, along with two hard negatives $\vec{d_j}$ and $\vec{d_k}$, where $\vec{d_j}$ is closer to the positive document $\vec{d_i}$, while $\vec{d_k}$ is far apart. Assuming a batch size of two, we illustrate how InfoNCE loss updates $\vec{q_i}$ when applied with the documents $\{\vec{d_i}, \vec{d_j}\}$ in panel (b) as opposed to $\{\vec{d_i}, \vec{d_k}\}$ in panel (c). Methods such as ANCE (Xiong et al., 2021), which select negatives based solely on the query–document similarity, tend to prefer negatives like $\vec{d_j}$. As shown in panel (b), such near-positive negatives induce only a weak improvement in the query's projection onto the positive direction $\vec{d_i}$. In contrast, panel (c) shows that the negatives preferred by HOBIT induce a substantially larger increase in projection toward $\vec{d_i}$. These findings underscore that informative negatives are not merely those that score highly with respect to the query, but those that also remain well separated from the positive.

# 5. **HOBIT** Solution Approach

Given a seed set $\mathcal{S} \subset D_{\text{trn}}$ with $|\mathcal{S}| = s < b$, the batch construction problem aims to select $b - s$ additional examples from the candidate pool $\mathcal{U} = D_{\text{trn}} \setminus \mathcal{S}$ to form a mini-batch $B = \mathcal{S} \cup X$. The objective is to choose $X$ so as to maximize the learning signal for queries in $\mathcal{S}$ by introducing hard yet non-contradictory in-batch negatives.

## 5.1. Batch Selection Problem

Now, we formally define the batch hardness measure.

**Definition 5.1** (Batch Hardness). For a candidate subset $X \subset \mathcal{U}$, we define the batch hardness as

$$H(B) = \sum_{q_i \in \mathcal{S}} \max_{(q_j, d_j) \in \mathcal{S} \cup X} w_{ij}, \tag{7}$$

where $B = X \cup \mathcal{S}$, and $w_{ij}$ is defined in Eq. 6.

The resulting batch selection problem is

$$\max_{X \subset \mathcal{U}} H(X \cup \mathcal{S}) \quad \text{subject to} \quad |X| = k. \tag{8}$$

where $k = b - s$.

**Proposition 5.2** (NP-hardness of Batch Hardness Maximization). *Maximizing the batch hardness objective in Eq. 8 is NP-hard.*
*We prove this via reduction to the Maximum Coverage problem, with the proof deferred to Appendix A.3.3.*

## 5.2. **HOBIT**'s Objective

HOBIT seeks a tractable batch selection objective that admits efficient optimization. The key idea is to replace the hard max in Eq. 7 using a smooth *Log-Sum-Exp* function, a standard smooth approximation to the maximum (Boyd & Vandenberghe, 2004):

$$\widetilde{h}_i(B) \; = \; \tau_h \log \left( \sum_{j \in B} \exp \left( \frac{w_{ij}}{\tau_h} \right) \right); \text{ for each } q_i \in \mathcal{S} \tag{9}$$

**Definition 5.3** (HOBIT's Objective). The batch hardness score optimized by HOBIT is defined as the sum of the smooth marginal contributions for the queries in the seed set:

$$\widetilde{H}(B) = \sum_{q_i \in \mathcal{S}} \widetilde{h}_i(B) = \tau_h \sum_{q_i \in \mathcal{S}} \log \left( \sum_{j \in B} \exp \left( \frac{w_{ij}}{\tau_h} \right) \right). \tag{10}$$

The optimization objective is $\max_{X \subset \mathcal{U}} \widetilde{H}(\mathcal{S} \cup X)$ s.t. $|X| = b - s$.

**Theorem 5.4** (Approximation Bounds). *Let $B = \mathcal{S} \cup X$ be the full batch of size $b$ with $|\mathcal{S}| = s$. For any real-valued weights $w_{ij}$, the HOBIT's batch hardness $\widetilde{H}(B)$ bounds the hardness objective $H(B)$ in Eq. 7 as follows:*

$$H(B) \; \leq \; \widetilde{H}(B) \; \leq \; H(B) + s\tau_h \log b, \tag{11}$$

*Proof deferred to Appendix A.4.1.* □

**Proposition 5.5** (Small-Temperature Regime). *As $\tau_h \to 0^+$, the surrogate objective converges to the sum of maximum interactions over the seed set:*

$$\lim_{\tau_h \to 0^+} \widetilde{H}(B) = \sum_{q_i \in \mathcal{S}} \max_{j \in B} w_{ij} = H(B). \quad (12)$$

As a consequence, we have this corollary.

**Corollary 5.6.** *For any two batches $B_1, B_2$, the ranking between $H$ and $\widetilde{H}$ is the same for $\tau_h \to 0^+$:*

$$\lim_{\tau_h \to 0^+} \left( \widetilde{H}(B_1) - \widetilde{H}(B_2) \right) = H(B_1) - H(B_2). \quad (13)$$

*Proof deferred to Appendix A.4.2.* □

**Proposition 5.7** (Large-Temperature Regime). *As $\tau_h \to \infty$, HOBIT's objective is asymptotically equivalent to:*

$$\lim_{\tau_h \to \infty} \widetilde{H}(B) = s\tau_h \log b + \mathcal{L}_{avg}(B). \quad (14)$$

*where $\mathcal{L}_{avg}(B) = \sum_{q_i \in \mathcal{S}} \left( \frac{1}{b} \sum_{j \in B} w_{ij} \right)$*

These results lead us to the following corollary.

**Corollary 5.8.** *For any two batches $B_1$ and $B_2$ of equal size, their ordering in the high-temperature limit is determined by the difference between their arithmetic-mean hardness scores:*

$$\lim_{\tau_h \to \infty} \left( \widetilde{H}(B_1) - \widetilde{H}(B_2) \right) = \left[ \mathcal{L}_{avg}(B_1) - \mathcal{L}_{avg}(B_2) \right]. \quad (15)$$

*Proof deferred to Appendix A.4.3.* □

*Remark* 5.9. Together, these two regimes show that $\widetilde{H}(B)$ adapts its selection behavior as a function of the batching temperature $\tau_h$. As $\tau_h \to 0$, the optimization emphasizes the **maximum** interaction, whereas as $\tau_h \to \infty$, it favors the **average** interaction. This behavior mirrors the role of the InfoNCE temperature $\tau$ in Eq. 2: at low temperatures, the gradient is dominated by the hardest negative, while at high temperatures, contributions from all negatives become more uniform. The two temperatures need not be tied; in our experiments, we set $\tau_h = \tau = 0.05$ to align batch construction with the hardness scale used by the training loss while avoiding an additional tuned hyperparameter.

### 5.3. Algorithm Design: Submodularity and Greedy Optimization

We now show that the objective $\widetilde{H}(B = \mathcal{S} \cup X)$ is monotone and submodular in $X$.

**Definition 5.10** (Marginal Gain). For a set $M \subseteq \mathcal{U}$ and an element $v \in \mathcal{U} \setminus M$, the marginal gain of adding $v$ to $M$ is

$$\Delta(v \mid M, \mathcal{S}) = \widetilde{H}(\mathcal{S} \cup M \cup \{v\}) - \widetilde{H}(\mathcal{S} \cup M)$$

$$= \sum_{q_i \in \mathcal{S}} \tau_h \log \left( 1 + \frac{\exp(w_{iv}/\tau_h)}{\sum_{j \in S \cup M} \exp(w_{ij}/\tau_h)} \right).$$
$$(16)$$

**Theorem 5.11** (Monotone submodularity). *The objective $\widetilde{H}(\mathcal{S} \cup X)$ is monotone and submodular with respect to the set $X \subseteq \mathcal{U}$.*

*Proof deferred to Appendix A.5.* □

**Corollary 5.12** (Greedy approximation guarantee). *Under a cardinality constraint $|B| = b$, the greedy algorithm achieves a $(1 - 1/e)$-approximation to the optimal value of $\widetilde{H}(B)$. (Nemhauser et al., 1978)*

### 5.4. **HOBIT**'s Algorithm

---
**Algorithm 1** HOBIT Training
---
1: **Input:** Training pairs $D_{trn} = \{(q_i, d_i)\}_{i=1}^N$, epochs $T$
2: **Input:** Batch parameters $b, s, k_{cand}$ and temperatures $\tau, \tau_h$
3: **Input:** Non-contradiction weight $\alpha$
4: **Return:** Trained dual encoder $f_\theta$
5: **for** $e = 1$ to $T$ **do**
6:    $E \leftarrow \{\vec{q}_i, \vec{d}_i\}_{i=1}^N$ using $f_{\theta^{e-1}}$ or the cache
7:    Set $P \leftarrow (D_{trn}, b, s, k_{cand}, \tau_h, \alpha, E)$
8:    $\mathcal{B}^{(e)} \leftarrow \text{SAMPLEBATCHES}(P)$
9:    **for** each batch $B \in \mathcal{B}^{(e)}$ **do**
10:      */* Forward training pass on batch B */*
11:      Compute loss: $\mathcal{L}(B) \leftarrow \text{INFONCE}(B, \tau)$ (Eq. 2)
12:      $\theta \leftarrow \text{GRADDESCENT}(\mathcal{L}(B), \theta)$
13:    **end for**
14: **end for**
---

An epoch of dual-encoder training comprises $t = |D_{trn}|/b$ SGD steps. At a given training step $k$, applying HOBIT requires computing hardness scores $w_{ij}$ (Eq. 6) using the current encoder parameters $f_{\theta^k}$. In an ideal implementation, query and document embeddings would be recomputed at every SGD step, and each batch would be drawn from examples not yet encountered within the epoch. Such a strategy, however, is computationally impractical, as the cost of batch construction alone would dominate the overall training time.

To make batch selection tractable, we instead consider a class of *lagged batch construction* strategies, in which batches are formed using embeddings produced by a lagged encoder $f_{\theta^{k-\ell_k}}$ with delay $\ell_k \geq 0$. This abstraction enables a principled analysis of our algorithms: HOBIT and

`HOBIT-C` as efficient, controlled approximations to the ideal step-wise batching implementation.

---

**Algorithm 2** SAMPLEBATCHES

---

1: **Input:** $D_{\text{trn}} = \{(q_i, d_i)\}_{i=1}^N$, $b$, $s$, $k_{\text{cand}}$, $\tau_h$, $\alpha$, $E = \{\vec{q_i}, \vec{d_i}\}_{i=1}^N$
2: **Return:** Ordered mini-batches $\mathcal{B}$
3: Initialize $\mathcal{U} \leftarrow D_{\text{trn}}, \mathcal{B} \leftarrow ()$
4: **while** $\mathcal{U} \neq \emptyset$ **do**
5:     Sample $\mathcal{S} \subseteq \mathcal{U}$ of size $s$ uniformly; $\mathcal{U} \leftarrow \mathcal{U} \setminus \mathcal{S}$
6:     Set $X \leftarrow \emptyset$ and collect the candidate pool

$$\mathcal{C} \leftarrow \bigcup_{q_i \in \mathcal{S}} \text{TOP-}k_{\text{cand}}(\vec{q_i}^\top \vec{d_j} : (q_j, d_j) \in \mathcal{U})$$

7:
8:     **while** $|X| < b - s$ and $\mathcal{C} \neq \emptyset$ **do**
9:         Select $v^\star$ that yields the highest marginal gain;

$$v^\star = \underset{v \in \mathcal{C}}{\arg\max} \, \Delta(v \mid X, \mathcal{S}) \text{ (Eq. 16)}$$

10:         $X \leftarrow X \cup \{v^\star\}, \mathcal{C} \leftarrow \mathcal{C} \setminus \{v^\star\}$
11:     **end while**
12:     $B \leftarrow \mathcal{S} \cup X, \mathcal{U} \leftarrow \mathcal{U} \setminus X, \mathcal{B} \leftarrow \text{append}(\mathcal{B}, B)$
13: **end while**
14: **return** $\mathcal{B}$

---

Here $k_{\text{cand}}$ controls the size of the top-$k$ candidate pool and is distinct from $\alpha$, which enters only through the hardness score $w_{ij}$ in Eq. 6.

**Our Algorithms.** We present two practical instantiations of our framework that differ in how the lag $\ell_k$ is handled.

- **HOBIT.** For all SGD steps within epoch $e$, batch construction uses embeddings computed by the encoder at the end of the previous epoch. Consequently, for any training step $k$ in epoch $e$, the lag satisfies $\ell_k \in \{0, 1, \ldots, t-1\}$.
- **HOBIT-C.** Batch construction relies on embeddings cached from forward passes performed during the earlier epoch $e-1$. This results in a rolling lag in which different examples may be associated with embeddings computed at different parameter states.

In Appendix A.6, we provide formal approximation guarantees for both algorithms. Under a mild *step-wise drift* assumption that: the embedding change per SGD step is bounded by a small $\epsilon$, as is typical with Adam or SGD with small learning rates, the error from using lagged embeddings grows linearly with the lag $\ell_k$. This highlights that in HOBIT, embeddings are synchronized at epoch boundaries, so the error depends on at most $\ell_k \leq t$; whereas in HOBIT-C, embeddings are drawn from a rolling cache spanning the previous epoch, yielding a worst-case error with $\ell_k \leq 2t$.

*Table 1.* Dataset statistics used in our experiments.

| Dataset | Train Pairs | Dev Pairs | Test Pairs | Corpus Size |
|---|---|---|---|---|
| MS MARCO (Passage) | 502,939 | 6,980 | – | 8,841,823 |
| Natural Questions | 58,880 | 8,757 | 3,610 | 21,015,324 |
| TriviaQA | 60,413 | 8,837 | 11,313 | 21,015,324 |
| NOMIC | 1,692,672 | – | – | – |

We provide the pseudocode for the dual encoder training using HOBIT in Alg. 1 and analyze its complexity next.

**Complexity analysis.** For constructing a batch of size $b$ with seed set size $s$ and candidate pool $\mathcal{C}$, lazy greedy maximization of the HOBIT's objective requires $\mathcal{O}(|\mathcal{C}| \log |\mathcal{C}|)$ marginal gain evaluations in the worst case. Each marginal gain computation costs $\mathcal{O}(s)$, as it aggregates contributions over the seed set. Therefore, the time complexity of lazy greedy batch construction is $\mathcal{O}(s|\mathcal{C}| \log |\mathcal{C}|)$.

## 6. Experiments and Results

Our experiments answer the following research questions:

**RQ1** Does our principled, training loss-aware batching outperform other state-of-the-art batching methods?
**RQ2** How well does HOBIT scale to large-scale retrieval training and benchmark evaluation?
**RQ3** Does HOBIT deliver complementary gains when combined with existing hard-negative mining methods?
**RQ4** How robust is HOBIT when q-d relevance labels are partially missing?
**RQ5** What is the computational overhead introduced by HOBIT and HOBIT-C?
**RQ6** How sensitive is HOBIT to $\tau_h$, $\alpha$, and $s$?

### 6.1. Experimental Setup

Our experiments follow standard dense retrieval protocols from recent work (Karpukhin et al., 2020; Xiong et al., 2021; Qu et al., 2021; Monath et al., 2023). Models are trained with the InfoNCE loss using in-batch negatives, with most hyperparameters (e.g., batch size and InfoNCE temperature $\tau$) set to the defaults from prior studies.

**Datasets.** Following prior work (Monath et al., 2023; Xiong et al., 2021; Yang et al., 2024), we evaluate our method on three standard dense retrieval benchmarks: MS MARCO Passage Ranking (Nguyen et al., 2016), Natural Questions (NQ) (Kwiatkowski et al., 2019), and TriviaQA (Joshi et al., 2017). Table 1 summarizes the dataset statistics.

**Encoders.** Table 2 shows the two dual-encoder backbones: `roberta-base` (Liu et al., 2019), a standard MLM-pretrained encoder, and `e5-large-unsupervised` (Wang et al., 2022), pretrained with weak supervision for retrieval. Queries and documents are encoded with a shared encoder using [CLS]

*Table 2.* Dual-encoder backbones used in our experiments.

| Model | Layers | Hidden Dim. | # Parameters |
|---|---|---|---|
| `roberta-base` | 12 | 768 | 125M |
| `e5-large-unsupervised` | 24 | 1024 | 334M |

*Table 3.* Comparison of state-of-the-art batching methods across datasets and encoder backbones. **Bold** highlights the best-performing method, and underlining marks the second-best.

| Method | MSMARCO Passage | | NQ | | TriviaQA | |
|---|---|---|---|---|---|---|
| | MRR@10 | MRR@100 | R@10 | R@100 | R@10 | R@100 |
| | | | `roberta-base` | | | |
| Random | 24.4 | 25.7 | 45.1 | 77.7 | 48.7 | 85.1 |
| NGAME | 26.8 | 28.1 | 47.9 | 78.9 | 51.5 | 86.9 |
| BatchSampler | 24.2 | 25.4 | 49.1 | 81.4 | 48.4 | 85.0 |
| B3 | 23.4 | 24.7 | 44.4 | 76.1 | 46.9 | 82.9 |
| HOBIT | **27.4** | **28.6** | **52.4** | **83.1** | **52.5** | **87.2** |
| | | | `e5-large-unsupervised` | | | |
| Random | 27.2 | 28.5 | 56.1 | 88.5 | 53.2 | 89.2 |
| NGAME | 28.6 | 29.9 | 61.3 | **90.5** | 56.0 | 90.1 |
| BatchSampler | 27.0 | 28.2 | 57.4 | 88.3 | 52.9 | 88.8 |
| B3 | 27.0 | 28.3 | 60.4 | 90.0 | 52.5 | 87.9 |
| HOBIT | **30.1** | **31.3** | **62.9** | 89.9 | **57.7** | **91.2** |

pooling and $\ell_2$ normalization. For E5, we use task-specific prompts (e.g., `query:`, `passage:`).

**Implementation Details.** We train all models using In-foNCE loss (Eq. 2). We use a batch size of 128 and set the softmax temperature to $\tau = 0.05$, following prior work (Hou & Li, 2023; Lei et al., 2023). We optimize using AdamW (Loshchilov & Hutter, 2019) with a learning rate of $1 \times 10^{-5}$, weight decay 0.01, gradient clipping at 1.0, and a linear warmup. We train models for up to 40 epochs with early stopping based on development-set performance (patience of 5 epochs). We use mixed-precision (FP16) training throughout. For HOBIT, we set the seed set size $s = 8$. For HOBIT, we set the batching temperature to $\tau_h = 0.05$, matching the InfoNCE temperature $\tau = 0.05$, and $\alpha$ defaults to 1. All experiments allow up to 64 positives per query. We conduct all experiments on a single NVIDIA B200 GPU.

**Evaluation Metrics.** At inference time, given a test query, the dual encoder retrieves relevant documents from the full corpus. We assess the performance using a comprehensive list of metrics: normalized Discounted Cumulative Gain@$k$ (nDCG@$k$), Recall@$k$ ($R@k$), and Mean Reciprocal Rank@$k$ (MRR@$k$) for $k \in \{10, 100\}$. For brevity, we report $R@k$ for the NQ, TriviaQA datasets, and $MRR@k$ for MSMARCO, following prior work. We defer the complete results to Appendix A.13, with formal definitions provided in Appendix A.7.

## 6.2. Baselines

We group the baselines into two categories:

**Batch Construction Methods.** *Random* uniformly shuffles training instances at each epoch. *NGAME* performs clustering over the query embeddings to group semantically similar queries within the same batch (Dahiya et al., 2023). *BS* constructs batches via graph-based random walks over query-query similarity graphs (Yang et al., 2023). *B3* forms semantically coherent batches using teacher-guided representations and graph partitioning (Thirukovalluru et al., 2025). We use `embeddinggemma-300m` (Vera et al., 2025) as the teacher model for B3.

**Hard-Negative Mining.** We consider two representative hard-negative mining methods, ANCE (Xiong et al., 2021) and TriSampler (Yang et al., 2024), to assess whether applying HOBIT atop mined negatives yields gains beyond training solely with mined negatives under random batching. ANCE maintains an approximate nearest-neighbor index over the document corpus and retrieves hard negatives by selecting high-scoring non-positive documents, e.g., $d^- = \arg\max_{d \in \mathcal{D} \setminus \{d^+\}} q^\top d$. Since rebuilding this index at every epoch is expensive, we refresh it periodically, using $R = 1$ for NQ and TriviaQA and $R = 2$ for MS MARCO (matching Xiong et al. (2021)). TriSampler first retrieves top-$K$ candidates and then samples negatives using a quasi-triangular criterion that favors informative negatives while avoiding candidates that are either trivial or too close to the positive document.

*Remark 6.1.* Since the training set $D_{\text{trn}}$ is small, HOBIT performs exact nearest-neighbor search entirely on the GPU. For the much larger document corpus, hard negatives are efficiently mined using FAISS (Johnson et al., 2019).

### 6.3. RQ1: Comparison with Batching Methods

We report the results for RQ1 in Table 3 and highlight the following observations:

1. Random batching performs significantly worse, showing the value of having informative in-batch negatives.
2. B3 underperforms because it keeps batches fixed and fails to account for encoder drift during training. In contrast, NGAME derives batch order each epoch, ranking second-best. This underscores that effective batching must adapt continuously during training to remain synchronized with the encoder's representation space.
3. With E5-large, HOBIT outperforms prior methods by a significant margin in all settings, except for R@100 on NQ, where it trails NGAME by just 0.6 points.
4. Overall, HOBIT emerges as the best performer.

### 6.4. RQ2: Large-Scale Retrieval Training and Benchmark Evaluation

We further evaluate whether HOBIT scales beyond the three supervised retrieval benchmarks above. Specifically, we

*Table 4.* Impact of combining HOBIT with explicit negative mining. All experiments use roberta-base with 8 sampled negatives per query. We report MRR@10/100 for MSMARCO Passage and Recall@10/100 for NQ and TriviaQA.

| Dataset | Method | MRR@10 | MRR@100 |
|---|---|---|---|
| MSMARCO | ANCE | **27.33** | **28.60** |
| | ANCE + HOBIT | 27.29 | 28.53 |
| | TriSampler | 26.26 | 27.46 |
| | TriSampler + HOBIT | **26.65** | **27.88** |
| | | R@10 | R@100 |
| NQ | ANCE | **59.27** | 84.63 |
| | ANCE + HOBIT | 56.43 | **85.41** |
| | TriSampler | 53.80 | 80.88 |
| | TriSampler + HOBIT | **57.13** | **84.75** |
| | | R@10 | R@100 |
| TriviaQA | ANCE | 52.67 | 86.34 |
| | ANCE + HOBIT | **54.46** | **88.34** |
| | TriSampler | 53.76 | 85.60 |
| | TriSampler + HOBIT | **54.11** | **87.75** |

fine-tune nomic-embed-text-v1-unsupervised for one epoch on the NOMIC training mixture, which contains 1,692,672 query-document pairs drawn from 10 source datasets, while changing only the batching strategy. We evaluate on MTEB retrieval tasks (Muennighoff et al., 2023). Since examples from different sources may not form valid negatives, we use source-aware batching: HOBIT is run independently within each source, and each batch combines four sources with 64 examples per source.

1. HOBIT improves 13 of 15 MTEB retrieval tasks over random batching.
2. The average gain is +2.88 nDCG@10, with a paired $t$-test across tasks showing statistical significance ($p = 0.001$).
3. The only exceptions are ArguAna and SciFact, where random batching is slightly better; detailed setup and per-task results are provided in Appendix A.8.

### 6.5. RQ3: Adding **HOBIT** to Hard-Negative Mining

Table 4 shows the results for RQ3.

1. *MSMARCO:* HOBIT yields limited gains when combined with ANCE and modest improvements with TriSampler, suggesting diminishing returns when strong mined negatives are already available.
2. *NQ:* While ANCE alone achieves the best Recall@10, combining it with HOBIT improves Recall@100, indicating benefits at larger cutoffs.
3. *TriviaQA:* HOBIT consistently improves performance when combined with both ANCE and TriSampler, achieving the best results across metrics.

Thus, when explicit hard negatives are available, enriching in-batch negatives either maintains performance or offers marginal gains, but with weaker negatives, HOBIT yields substantial gains. Therefore, HOBIT complements rather than replaces hard-negative mining.

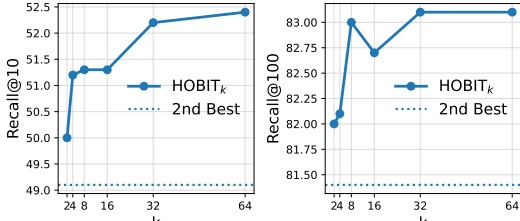

*Figure 3.* Effect of missing relevance labels on $\text{HOBIT}_k$ for NQ. Recall@10 (left) and Recall@100 (right) are shown, with dotted lines marking the strongest non-HOBIT batch sampler.

*Table 5.* Wall-clock overhead on roberta-base. Forward/backward time is common within each dataset; overhead denotes method-specific preprocessing per epoch. Scores are R@10 for NQ and MRR@10 for MSMARCO.

| Dataset | Method | Fwd./Bwd. | Overhead | OH % | Score |
|---|---|---|---|---|---|
| NQ | Random | 155s | 0s | 0% | 45.1 |
| | NGAME | 155s | 12s | 8% | 47.9 |
| | HOBIT | 155s | 35s | 23% | **52.4** |
| | HOBIT-C | 155s | 0.2s | 0.1% | 51.7 |
| MSMARCO | Random | 460s | 0s | 0% | 24.4 |
| | NGAME | 460s | 65s | 14% | 26.8 |
| | HOBIT | 460s | 130s | 28% | **27.4** |
| | HOBIT-C | 460s | 8s | 1.7% | 27.2 |

### 6.6. RQ4: Impact of Missing Labels on **HOBIT**

We evaluate HOBIT's robustness to sparse relevance supervision by limiting the number of positives per query ($k \in \{2, 4, 8, 16, 32, 64\}$). All experiments use roberta-base. We excluded MSMARCO Passage since each query has only one ground-truth label. Results for NQ are shown in Fig. 3, and for TriviaQA in Fig. 5 in the Appendix.

1. HOBIT degrades gracefully as supervision becomes sparse and remains competitive even in the most restrictive setting ($k = 2$).
2. Performance improves monotonically with increasing $k$, and HOBIT consistently outperforms the strongest baseline for moderate sparsity levels ($k \geq 8$).
3. This robustness is attributed to the hardness scoring in HOBIT, where the non-contradiction term downweights false-negative-like candidates during batch construction.

### 6.7. RQ5: Efficiency and Cached Batch Construction

We compare the computational efficiency of HOBIT with HOBIT-C in Table 6, which reports per-epoch speedups. Table 5 further breaks down wall-clock cost into common forward/backward time and method-specific preprocessing overhead.

1. HOBIT-C achieves substantial speedups across datasets (14–30%) with only minor drops in performance.

*Table 6.* Comparison of `HOBIT` vs. cached `HOBIT-C`. Efficiency gain shows per-epoch speedup of `HOBIT-C` over `HOBIT`. All settings use `roberta-base` encoder.

| Dataset | Method | MRR@10 | MRR@100 | Efficiency |
|---------|--------|--------|---------|------------|
| MSMARCO | HOBIT | 27.4 | 28.6 | – |
|  | HOBIT-C | 27.2 | 28.3 | **14.3%** |
|  |  | R@10 | R@100 | Efficiency |
| NQ | HOBIT | 52.4 | 83.1 | – |
|  | HOBIT-C | 51.7 | 82.4 | **29.4%** |
|  |  | R@10 | R@100 | Efficiency |
| TriviaQA | HOBIT | 52.5 | 87.2 | – |
|  | HOBIT-C | 52.2 | 86.9 | **21.4%** |

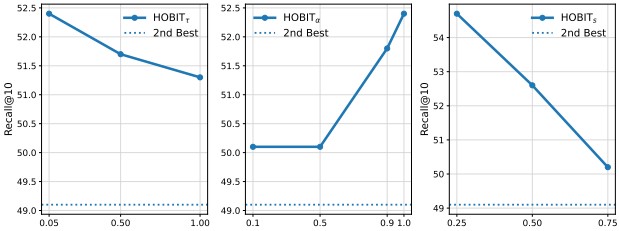

*Figure 4.* `HOBIT` sensitivity to $\tau_h$, $\alpha$, and $s$.

2. `HOBIT-C` adds only 0.2s overhead per epoch on NQ and 8s on MSMARCO by reusing cached embeddings, while `HOBIT` achieves the best absolute retrieval quality at higher preprocessing cost.
3. The performance drop remains less than one point across all metrics, showing that caching robustly preserves the benefits of batch construction, which is also in line with Theorem A.5.
4. Overall, caching significantly reduces training time, making batch construction practical for large-scale datasets.

### 6.8. RQ6: Sensitivity to $\tau_h$, $\alpha$, and $s$

We study the sensitivity of `HOBIT` to three hyperparameters: batching temperature $\tau_h$ used in Eq. 10, the mixing coefficient $\alpha$ used in $w_{ij}$ (Eq. 6), and the seed set size $s$ used to initialize each batch. We plot the results in Fig. 4.

1. $\tau_h$: Performance peaks at the default $\tau_h = 0.05$ and declines gradually with higher $\tau_h$, yet stays competitive with the strongest non-HOBIT sampler. This emphasizes the need to keep batch construction aligned with the hardness scale of the underlying training loss when selecting negatives.
2. $\alpha$: Performance rises steadily with $\alpha$, peaking at 1.0, underscoring the importance of discarding near-positive hard negatives, a step overlooked by most baselines. We also tested linearly annealing $\alpha$ from 0 to 1, which achieved 52.2 R@10 versus 52.4 for fixed $\alpha = 1.0$, indicating that HOBIT exhibits robustness even under such annealing schedules on $\alpha$.
3. $s$: We vary the seed set size as a fraction of the batch and find that smaller seeds improve retrieval performance, as

they give `HOBIT` more flexibility to enrich batches with diverse, informative hard negatives, whereas larger seeds limit the scope for batch refinement. Appendix A.11 shows that alternative seed initialization strategies provide similar retrieval performance.

### 6.9. Additional Experiments

We include additional ablations in the appendix to validate the robustness of the main findings.

- **Warmup and staleness.** Appendix A.9 shows that `HOBIT`'s performance does not degrade without a warmup phase, and further validates `HOBIT`'s efficiency.
- **Batch-size sensitivity.** Appendix A.10 shows that `HOBIT` maintains a 6.8–7.5 R@10 advantage across batch sizes from 64 to 512.
- **Seed initialization.** Appendix A.11 compares random seeds with K-means and facility-location alternatives, showing similar retrieval quality across seed-selection strategies.

## 7. Limitations

`HOBIT` improves in-batch negative quality, but can increase end-to-end wall-clock time relative to random batching because harder batches may require more epochs before early stopping; `HOBIT-C` mitigates this cost by reusing cached embeddings at a small cost in final accuracy. Batch construction incurs an additional overhead over the per-epoch training loop and further hard batches may take longer for convergence.

## 8. Conclusion

We introduced `HOBIT`, a principled approach to dual encoder training that constructs informative mini-batches. We identified two complementary criteria for an effective in-batch negative: it should be hard for the query and non-contradictory with the positive. We derived the batch order by optimizing a monotone submodular objective via a greedy algorithm with a $(1 - 1/e)$ guarantee. Our experiments on MSMARCO, NQ, and TriviaQA showed that `HOBIT` consistently outperformed state-of-the-art batch samplers, improving MRR and Recall by 1–3 points. For datasets with mined hard negatives, we showed that our method improved performance in some settings and maintained it in others, thereby complementing rather than replacing them. We proposed `HOBIT-C` as a scalable variant that reduced per-epoch training time by 12–30% with less than a 1-point performance drop. Overall, our work shows that having loss-aware mini-batches with informative in-batch negatives can substantially boost retrieval performance while adding minimal overhead, making them highly effective and practical for large-scale retrieval systems.

## Impact Statement

This work introduces a lightweight, principled batching strategy for contrastive training of dense retrieval models, improving training stability and efficiency while reducing reliance on costly hard-negative mining. It is broadly applicable to retrieval and representation learning, and poses no direct societal risks.

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

# A. Appendix

## A.1. Qualitative Analysis of In-Batch Negatives

To complement our theoretical analysis and quantitative evaluation, we present a qualitative example illustrating how different batch sampling strategies populate in-batch negatives for the same query–positive pair at different stages of training. The example is drawn from the MS MARCO Passage dataset and is intended to highlight systematic differences in the *type* of negatives selected by each method, rather than to provide exhaustive coverage.

We consider a fixed query and its associated positive passage, and inspect representative in-batch negatives produced by three batching strategies: (i) random batching, (ii) query–document similarity-based batching (e.g., ANCE-style), and (iii) HOBIT. For each method, we show examples corresponding to early training and to mid/late training, reflecting the evolution of the embedding space.

*Table 7.* Example of in-batch negatives selected by different batching strategies for a fixed MS MARCO query–positive pair at different stages of training.

| Query–Positive Pair | | |
| --- | --- | --- |
| **Query:** When is the best time to travel to Iceland for the northern lights? | | |
| **Positive Document:** In Iceland—especially far up in the north—it is possible to see the beautiful Aurora. The best time is from October to April each year, when we have clear skies and cold weather. It is ideal due to the fact that Iceland lies between two continents and has many excellent locations for viewing the Northern Lights. | | |
| **Batching Method** | **Early Epoch In-Batch Negative** | **Mid / Late Epoch In-Batch Negative** |
| **Random (R)** | *Will dental insurance cover surgery* \| Dental insurance policies typically cover procedures related to teeth and oral health. Medical insurance generally does not cover dental surgery, though coverage varies by policy. | *Federal court system definition* \| A federal court is a court established by the authority of a federal government. The federal judiciary of the United States is responsible for interpreting and enforcing federal laws. |
| **ANCE-style (A)** | *Are the northern lights seasonal* \| Prime seasons for viewing the northern lights are early autumn and spring, with visibility typically between September and March depending on solar activity. | *When do the northern lights appear in Iceland?* \| The best time of year to see the Northern Lights in Iceland is between September and March, with clear skies and cold but dry conditions offering the best chance. |
| **HOBIT (HOBIT)** | *When can we see northern lights in Norway* \| The Northern Lights usually appear between 6 p.m. and 1 a.m., with peak frequency around 10–11 p.m. during the dark months. | *How often do the northern lights appear* \| The Northern Lights occur year round but are only visible when nights are dark. In regions such as Alaska, Canada, Norway, and Finland, they are most commonly seen from September through April. |

This example illustrates that while ANCE-style batching increasingly selects negatives that are lexically and semantically very close to the positive passage, such negatives can be ambiguous and partially contradictory. In contrast, HOBIT consistently selects negatives that remain highly relevant to the query while avoiding near-duplicates of the positive, aligning with the design of the hardness score in Eq. 6.

## A.2. Detailed Related Work

We provide a comprehensive review of the related literature on dual encoder training and hard-negative mining approaches below:

### A.2.1. DENSE RETRIEVAL AND CONTRASTIVE OBJECTIVES.

Dual-encoder dense retrieval trained with contrastive objectives such as InfoNCE is the dominant paradigm for scalable retrieval (Karpukhin et al., 2020; van den Oord et al., 2018). In this setting, in-batch negatives serve as an efficient approximation to global negatives, but their effectiveness depends critically on batch composition (Karpukhin et al., 2020; Lin et al., 2021a; Wang & Isola, 2020), as the negative distribution in the softmax denominator directly shapes gradient signal and embedding geometry (Cao et al., 2022). Retrieval performance is further improved through better pre-training and objective design: domain-matched and corpus-aware pre-training yield stronger encoder initializations (Oguz et al., 2022; Gao & Callan, 2021; 2022; Wang et al., 2023; Shen et al., 2023), while weakly supervised and unsupervised contrastive objectives produce high-quality retrieval embeddings (Wang et al., 2022; Izacard et al., 2022; Gupta et al., 2024a). Several

methods enhance robustness by modifying the contrastive loss while still relying on in-batch negatives (Li et al., 2021; Lu et al., 2022). To increase the effective number of negatives without prohibitive cost, momentum queues and memory banks reuse past embeddings (He et al., 2020; Chen et al., 2020), and retrieval-specific approximations such as negative caches and gradient accumulation improve convergence under hardware constraints (Lindgren et al., 2021; Kim et al., 2024; Lin et al., 2021b).

### A.2.2. HARD-NEGATIVE MINING.

Hard-negative mining improves dense retrieval by retrieving challenging negatives from the corpus (Xiong et al., 2021; Yang et al., 2024; Lai et al., 2024; Shi et al., 2022; Chen et al., 2024; Monath et al., 2023). ANCE introduced asynchronous ANN-based mining, showing substantial gains over random or lexical BM25 negatives (Xiong et al., 2021), and subsequent systems expand the negative pool and reduce false negatives through cross-batch sharing and cross-encoder filtering (Qu et al., 2021; Ren et al., 2021b). Despite their effectiveness, these approaches are computationally expensive and prone to representation staleness due to delayed index updates (Xiong et al., 2021; Sun et al., 2022). Parallel work studies which negatives are most informative: SimANS shows that excessively hard negatives can introduce bias through false positives and proposes sampling ambiguous negatives (Zhou et al., 2022), while TriSampler enforces geometric constraints to avoid both trivial and false-negative-like samples (Yang et al., 2024). More broadly, noise-aware contrastive learning identifies false negatives as a central failure mode under sparse supervision (Chuang et al., 2020; Zhang & Stratos, 2021; Shi et al., 2023; Liu & Wang, 2023), motivating system-level remedies using auxiliary rankers and adversarial objectives (Zhang et al., 2022; Qu et al., 2021; Ren et al., 2021a).

### A.3. Proofs from Problem Formulation

#### A.3.1. PROOF OF LEMMA 4.1

*Proof.* Recall that the loss for the $i$-th item in the batch is

$$\mathcal{L}_i = -\frac{s_{ii}}{\tau} + \log\left(\sum_{j=1}^{b} \exp\left(\frac{s_{ij}}{\tau}\right)\right). \tag{17}$$

Since $s_{ij} = \vec{q}_i^{\top} \vec{d}_j$, we have $\nabla_{q_i} s_{ij} = \vec{d}_j$. Differentiating the two terms yields

$$\nabla_{\vec{q}_i}\left(-\frac{s_{ii}}{\tau}\right) = -\frac{1}{\tau}\vec{d}_i, \tag{18}$$

$$\nabla_{\vec{q}_i}\log\left(\sum_j \exp\left(\frac{s_{ij}}{\tau}\right)\right) = \frac{1}{\tau}\sum_j P_{ij}\,\vec{d}_j. \tag{19}$$

Combining the two expressions gives the result. $\square$

#### A.3.2. PROOF OF THEOREM 4.3

*Proof.* From the gradient expression for the InfoNCE loss,

$$\nabla_{\vec{q}_i}\mathcal{L}_i = \frac{1}{\tau}\left(\sum_{k=1}^{b} P_{ik}\,\vec{d}_k - \vec{d}_i\right). \tag{20}$$

We first expand the softmax-weighted document embedding:

$$\sum_{k=1}^{b} P_{ik}\,\vec{d}_k = (1 - p - \epsilon)\vec{d}_i + p\vec{d}_j + \sum_{k \notin \{i,j\}} P_{ik}\,\vec{d}_k. \tag{21}$$

Subtracting $\vec{d}_i$ from both sides yields

$$\sum_{k=1}^{b} P_{ik}\,\vec{d}_k - \vec{d}_i = (1 - p - \epsilon)\vec{d}_i - \vec{d}_i + p\vec{d}_j + \sum_{k \notin \{i,j\}} P_{ik}\,\vec{d}_k \tag{22}$$

$$= -(p + \epsilon)\vec{d}_i + p\vec{d}_j + \sum_{k \notin \{i,j\}} P_{ik}\,\vec{d}_k. \tag{23}$$

Now we regroup the terms to obtain:

$$- (p + \epsilon)\vec{d}_i + p\vec{d}_j + \sum_{k \notin \{i,j\}} P_{ik}\,\vec{d}_k \tag{24}$$

$$= p(\vec{d}_j - \vec{d}_i) + \sum_{k \notin \{i,j\}} P_{ik}(\vec{d}_k - \vec{d}_i), \tag{25}$$

where we used the identity

$$\sum_{k \notin \{i,j\}} P_{ik}\vec{d}_i = \epsilon\,\vec{d}_i. \tag{26}$$

Define

$$a = p(\vec{d}_j - \vec{d}_i), \qquad r = \sum_{k \notin \{i,j\}} P_{ik}(\vec{d}_k - \vec{d}_i). \tag{27}$$

Then

$$\sum_{k=1}^{b} P_{ik}\vec{d}_k - \vec{d}_i = a + r. \tag{28}$$

By the reverse triangle inequality,

$$\|a + r\|_2 \geq \|a\|_2 - \|r\|_2. \tag{29}$$

For the dominant term,

$$\|a\|_2 = p\|\vec{d}_j - \vec{d}_i\|_2 \tag{30}$$

$$= p\sqrt{\|\vec{d}_j\|_2^2 + \|\vec{d}_i\|_2^2 - 2\,\vec{d}_i^{\top}\vec{d}_j} \tag{31}$$

$$= p\sqrt{2(1 - t_{ij})}, \tag{32}$$

where $t_{ij} = \vec{d}_i^{\top}\vec{d}_j$ and embeddings are unit-normalized.

For the residual term, using the triangle inequality and $\|\vec{d}_k - \vec{d}_i\|_2 \leq 2$,

$$\|r\|_2 \leq \sum_{k \notin \{i,j\}} P_{ik}\,\|\vec{d}_k - \vec{d}_i\|_2 \tag{33}$$

$$\leq 2 \sum_{k \notin \{i,j\}} P_{ik} = 2\epsilon. \tag{34}$$

Combining the bounds,

$$\left\|\sum_{k=1}^{b} P_{ik}\vec{d}_k - \vec{d}_i\right\|_2 \geq p\sqrt{2(1 - t_{ij})} - 2\epsilon. \tag{35}$$

Dividing by $\tau$ gives

$$\left\|\nabla_{\bar{q}_i}\mathcal{L}_i\right\|_2 \geq \frac{p}{\tau}\sqrt{2(1-t_{ij})} - \frac{2\epsilon}{\tau}, \tag{36}$$

which completes the proof. $\square$

### A.3.3. PROOF OF PROPOSITION 5.2

*Proof.* We prove NP-hardness via a polynomial-time reduction from the classical MAXIMUM COVERAGE problem, which is known to be NP-hard.

An instance of MAXIMUM COVERAGE consists of a finite universe $U = \{u_1, \ldots, u_n\}$, a collection of subsets $\mathcal{C} = \{S_1, \ldots, S_m\}$ with $S_j \subseteq U$, and an integer $k$. The objective is to select $k$ subsets whose union has maximum cardinality.

Given such an instance, we construct an instance of the batch selection problem as follows. For each element $u_i \in U$, we create a distinct seed example $(q_i, d_i)$ and include it in the seed set $\mathcal{S}$. For each subset $S_j \in \mathcal{C}$, we create a distinct candidate example $(q_j, d_j)$ and include it in the candidate pool $\mathcal{U}$.

We define the interaction weights $w_{ij}$ synthetically to encode membership of elements in subsets:

$$w_{ij} = \begin{cases} 1, & \text{if } u_i \in S_j, \\ 0, & \text{otherwise.} \end{cases} \tag{37}$$

Additionally, we set all interactions among seed examples to zero, so that no seed example contributes hardness to another seed example.

Under this construction, for any subset $X \subset \mathcal{U}$, the inner maximization $\max_{(q_j,d_j) \in \mathcal{S} \cup X} w_{ij}$ evaluates to 1 if and only if there exists at least one candidate $(q_j, d_j) \in X$ such that $u_i \in S_j$, and evaluates to 0 otherwise. Consequently, the batch hardness reduces to

$$H(X) = \sum_{q_i \in \mathcal{S}} \mathbb{I}[\exists(q_j, d_j) \in X \text{ such that } u_i \in S_j] \tag{38}$$

$$= \left| \bigcup_{(q_j,d_j) \in X} S_j \right|.$$

Thus, maximizing $H(X)$ subject to $|X| = k$ is equivalent to selecting $k$ subsets from $\mathcal{C}$ whose union has maximum size, which is exactly the MAXIMUM COVERAGE problem. Since MAXIMUM COVERAGE is NP-hard, the batch hardness maximization problem is also NP-hard. $\square$

### A.4. Properties of the Surrogate Batch Hardness Objective

#### A.4.1. APPROXIMATION BOUNDS FOR LOG-SUM-EXP

*Proof.* We analyze the bounds for a single query term $\widetilde{h}_i(B)$ for a fixed $q_i \in \mathcal{S}$ and then sum over the seed set. Let $w_i^\star = \max_{j \in B} w_{ij}$.

**Lower bound.** Since the exponential function maps real inputs to positive values, $\exp(w_{ij}/\tau_h) > 0$ for all $j$. The sum of positive terms is strictly greater than or equal to its largest term:

$$\sum_{j \in B} \exp\left(\frac{w_{ij}}{\tau_h}\right) \geq \max_{j \in B}\left[\exp\left(\frac{w_{ij}}{\tau_h}\right)\right] = \exp\left(\frac{w_i^\star}{\tau_h}\right). \tag{39}$$

Taking the logarithm (which is monotonic) and multiplying by $\tau_h > 0$:

$$\tau_h \log\left(\sum_{j \in B} \exp\left(\frac{w_{ij}}{\tau_h}\right)\right) \geq \tau_h \log\left(\exp\left(\frac{w_i^\star}{\tau_h}\right)\right) = w_i^\star. \tag{40}$$

Thus, $\widetilde{h}_i(B) \geq \max_{j \in B} w_{ij}$. Summing over all $q_i \in \mathcal{S}$ yields $\widetilde{H}(B) \geq H(B)$.

**Upper bound.** For every $j \in B$, we have $\exp(w_{ij}/\tau_h) \leq \exp(w_i^\star/\tau_h)$. We can bound the sum by replacing every term with the maximum term:

$$\sum_{j \in B} \exp\left(\frac{w_{ij}}{\tau_h}\right) \leq \sum_{j \in B} \exp\left(\frac{w_i^\star}{\tau_h}\right) = b \cdot \exp\left(\frac{w_i^\star}{\tau_h}\right). \tag{41}$$

Taking the logarithm and multiplying by $\tau_h$:

$$\widetilde{h}_i(B) \leq \tau_h \log\left(b \cdot \exp\left(\frac{w_i^\star}{\tau_h}\right)\right) \tag{42}$$

$$= \tau_h\left(\log b + \frac{w_i^\star}{\tau_h}\right) \tag{43}$$

$$= w_i^\star + \tau_h \log b. \tag{44}$$

Summing over all $q_i \in \mathcal{S}$ yields:

$$\sum_{q_i \in \mathcal{S}} \widetilde{h}_i(B) \leq \sum_{q_i \in \mathcal{S}} \left(\max_{j \in B} w_{ij} + \tau_h \log b\right) \tag{45}$$

$$= H(B) + s\tau_h \log b.$$

$\square$

### A.4.2. SMALL-TEMPERATURE LIMIT

*Proof.* Let $w_i^\star = \max_{j \in B} w_{ij}$. We analyze the term for a single query $q_i$:

$$\widetilde{h}_i(B) = \tau_h \log\left(\sum_{j \in B} \exp\left(\frac{w_{ij}}{\tau_h}\right)\right) \tag{46}$$

$$= \tau_h \log\left(\exp\left(\frac{w_i^\star}{\tau_h}\right) \sum_{j \in B} \exp\left(\frac{w_{ij} - w_i^\star}{\tau_h}\right)\right) \tag{47}$$

$$= w_i^\star + \tau_h \log\left(\sum_{j \in B} \exp\left(\frac{w_{ij} - w_i^\star}{\tau_h}\right)\right). \tag{48}$$

Let $n_i$ be the number of items in $B$ that achieve the maximum score $w_i^\star$. For any $j$ where $w_{ij} < w_i^\star$, the term $\exp((w_{ij} - w_i^\star)/\tau_h)$ approaches 0 as $\tau_h \to 0^+$. The sum inside the logarithm, therefore, approaches $n_i$.

$$\lim_{\tau_h \to 0^+} \widetilde{h}_i(B) = w_i^\star + \lim_{\tau_h \to 0^+} \tau_h \log(n_i) = w_i^\star. \tag{49}$$

Summing over all $q_i \in \mathcal{S}$ yields the result. $\square$

**Corollary A.1.** *For any two batches $B_1, B_2$, the low-temperature difference between $H$ and $\widetilde{H}$ converges as follows:*

$$\lim_{\tau_h \to 0^+} \left(\widetilde{H}(B_1) - \widetilde{H}(B_2)\right) = H(B_1) - H(B_2). \tag{50}$$

*Proof.* Let $L = \lim_{\tau_h \to 0^+}(\widetilde{H}(B_1) - \widetilde{H}(B_2))$. By the linearity of limits:

$$L = \sum_{q_i \in \mathcal{S}} \lim_{\tau_h \to 0^+} \widetilde{h}_i(B_1) - \sum_{q_i \in \mathcal{S}} \lim_{\tau_h \to 0^+} \widetilde{h}_i(B_2). \tag{51}$$

Substituting the pointwise limit from Proposition 5.5 yields the stated result. $\square$

A.4.3. LARGE-TEMPERATURE LIMIT

*Proof.* We use the Taylor expansions $e^x = 1 + x + \mathcal{O}(x^2)$ and $\log(1+x) = x + \mathcal{O}(x^2)$ for small $x$. As $\tau_h \to \infty$, $w_{ij}/\tau_h \to 0$. Analyzing the inner sum for a single query $q_i$:

$$\sum_{j \in B} \exp\left(\frac{w_{ij}}{\tau_h}\right) = \sum_{j \in B} \left(1 + \frac{w_{ij}}{\tau_h} + \mathcal{O}(\tau_h^{-2})\right) \tag{52}$$

$$= b + \frac{1}{\tau_h} \sum_{j \in B} w_{ij} + \mathcal{O}(\tau_h^{-2}) \tag{53}$$

$$= b \left(1 + \frac{1}{b\tau_h} \sum_{j \in B} w_{ij} + \mathcal{O}(\tau_h^{-2})\right). \tag{54}$$

Substituting this back into the expression for $\widetilde{h}_i(B)$:

$$\widetilde{h}_i(B) = \tau_h \log\left[b\left(1 + \frac{1}{b\tau_h} \sum_{j \in B} w_{ij}\right)\right] \tag{55}$$

$$= \tau_h \log b + \tau_h \log\left(1 + \frac{1}{b\tau_h} \sum_{j \in B} w_{ij}\right) \tag{56}$$

$$= \tau_h \log b + \tau_h \left(\frac{1}{b\tau_h} \sum_{j \in B} w_{ij} + \mathcal{O}(\tau_h^{-2})\right) \tag{57}$$

$$= \tau_h \log b + \frac{1}{b} \sum_{j \in B} w_{ij} + \mathcal{O}(\tau_h^{-1}). \tag{58}$$

Summing over all $q_i \in \mathcal{S}$ yields the proposition. $\square$

**Corollary A.2.** *For any two batches $B_1, B_2$ of equal size $|B_1| = |B_2| = b$, the ranking in the high-temperature limit is determined by the difference of the arithmetic-mean hardness scores:*

$$\lim_{\tau_h \to \infty} \left(\widetilde{H}(B_1) - \widetilde{H}(B_2)\right) = \mathcal{L}_{avg}(B_1) - \mathcal{L}_{avg}(B_2). \tag{59}$$

*where $\mathcal{L}_{avg}(B) = \sum_{q_i \in \mathcal{S}} \frac{1}{b} \sum_{j \in B} w_{ij}$.*

*Proof.* Using the expansion from Proposition 5.7, we write the difference:

$$\widetilde{H}(B_1) - \widetilde{H}(B_2) = \tag{60}$$
$$(|\mathcal{S}|\tau_h \log b + \mathcal{L}_{avg}(B_1)) - (|\mathcal{S}|\tau_h \log b + \mathcal{L}_{avg}(B_2))$$

The divergent term $|\mathcal{S}|\tau_h \log b$ cancels out exactly because the batch sizes are equal. Taking the limit $\tau_h \to \infty$ removes the $\mathcal{O}(\tau_h^{-1})$ remainder, leaving only the difference of the means. $\square$

## A.5. HOBIT: Proof of Monotone Submodularity

*Proof.* We first show monotonicity. For any $N \subseteq \mathcal{U}$ and any $v \in \mathcal{U} \setminus N$, adding $v$ increases the argument of the logarithm for each $q_i \in \mathcal{S}$, since all exponential terms are non-negative. Therefore, $\widetilde{H}(\mathcal{S} \cup N \cup \{v\}) \geq \widetilde{H}(\mathcal{S} \cup N)$, and the objective is monotone.

We now prove submodularity via the diminishing-returns property. Consider two sets $M \subseteq N \subseteq \mathcal{U}$ and an element $v \in \mathcal{U} \setminus N$. For a fixed query $q_i \in \mathcal{S}$, define

$$Z_i(M) = \sum_{j \in \mathcal{S} \cup M} \exp(w_{ij}/\tau_h), \qquad Z_i(N) = \sum_{j \in \mathcal{S} \cup N} \exp(w_{ij}/\tau_h).$$

Since $M \subseteq N$, we have $Z_i(N) \geq Z_i(M)$.

From Eq. (16), the per-query marginal contribution of $v$ when added to a set $M$ is

$$\Delta_i(v \mid M) = \tau_h \log\left(1 + \frac{\exp(w_{iv}/\tau_h)}{Z_i(M)}\right).$$

The function $g(x) = \tau_h \log(1 + c/x)$ for fixed $c > 0$ is non-increasing in $x > 0$. Therefore, increasing the denominator from $Z_i(M)$ to $Z_i(N)$ can only decrease the marginal gain, implying

$$\Delta_i(v \mid M) \geq \Delta_i(v \mid N).$$

Summing this inequality over all $q_i \in \mathcal{S}$ yields

$$\Delta(v \mid M, \mathcal{S}) = \sum_{q_i \in \mathcal{S}} \Delta_i(v \mid M) \geq \sum_{q_i \in \mathcal{S}} \Delta_i(v \mid N) = \Delta(v \mid N, \mathcal{S}),$$

which establishes the diminishing-returns property. Hence, $\widetilde{H}(\mathcal{S} \cup N)$ is submodular in $N$. $\quad\square$

### A.6. Lagged Batch Construction: Formal Guarantees

In this appendix, we provide formal guarantees for lagged batch construction. We introduce a *step-wise drift assumption* and derive how this drift accumulates over the lag period $\ell_k$, explicitly linking the lag duration to the approximation bounds.

#### A.6.1. STEP-WISE EMBEDDING DRIFT

We assume that the embedding parameters change smoothly at each SGD step. This is consistent with standard optimization (e.g., SGD, Adam), where the learning rate limits the magnitude of parameter updates per step.

**Assumption A.3** (Bounded step-wise drift). There exists a constant $\epsilon > 0$ such that for any optimization step $k$ and all indices $i$, the change in embeddings between consecutive steps is bounded by:

$$\|\vec{q_i}^{(k)} - \vec{q_i}^{(k-1)}\|_2 \leq \epsilon, \qquad \|\vec{d_i}^{(k)} - \vec{d_i}^{(k-1)}\|_2 \leq \epsilon. \tag{61}$$

#### A.6.2. ACCUMULATED DRIFT UNDER LAG

We first establish that the total embedding drift for a lag of $\ell_k$ is bounded linearly by the number of lagged steps.

**Lemma A.4** (Lag-dependent drift bound). *Under Assumption A.3, for a current step $k$ and a lag $\ell_k \geq 0$, the total embedding drift is bounded by:*

$$\|\vec{q_i}^{(k)} - \vec{q_i}^{(k-\ell_k)}\|_2 \leq \ell_k\epsilon, \qquad \|\vec{d_i}^{(k)} - \vec{d_i}^{(k-\ell_k)}\|_2 \leq \ell_k\epsilon. \tag{62}$$

*Proof.* We express the difference between the current and lagged embedding as a telescoping sum of single-step updates. Using the triangle inequality and Assumption A.3:

$$
\begin{aligned}
\|\vec{q_i}^{(k)} - \vec{q_i}^{(k-\ell_k)}\|_2 &= \left\|\sum_{j=0}^{\ell_k-1} \left(\vec{q_i}^{(k-j)} - \vec{q_i}^{(k-j-1)}\right)\right\|_2 \\
&\leq \sum_{j=0}^{\ell_k-1} \left\|\vec{q_i}^{(k-j)} - \vec{q_i}^{(k-j-1)}\right\|_2 \\
&\leq \sum_{j=0}^{\ell_k-1} \epsilon = \ell_k\epsilon.
\end{aligned}
\tag{63}
$$

The derivation for $\vec{d_i}$ is identical. $\quad\square$

### A.6.3. STABILITY UNDER LAG

**Lemma A.5** (Lag-dependent perturbation). *Under Assumption A.3, for any fixed batch $B$ and lag $\ell_k$,*

$$\left| \widetilde{H}_{\theta_k}(B) - \widetilde{H}_{\theta_{k-\ell_k}}(B) \right| \le b(2 + 2\alpha)\ell_k\epsilon. \tag{64}$$

*Proof.* First, we bound the drift in the hardness scores $w_{ij}$. For unit-norm embeddings, the difference in dot products satisfies $|u^\top v - u'^\top v'| \le \|u - u'\| + \|v - v'\|$. Combining this with Lemma A.4:

$$|w_{ij}^{(k)} - w_{ij}^{(k-\ell_k)}| \le 2\ell_k\epsilon + 2\alpha\ell_k\epsilon = (2 + 2\alpha)\ell_k\epsilon. \tag{65}$$

The function $\tau_h \log \sum_{j \in B} \exp(w_{ij}/\tau_h)$ is 1-Lipschitz with respect to the scores $w_{ij}$. Summing over the batch of size $b$ yields the stated bound. $\square$

### A.6.4. GREEDY OPTIMIZATION UNDER LAG

**Theorem A.6** (Greedy guarantee with lag-dependent error). *Let $B_k^\star$ be an optimal batch for $\widetilde{H}_{\theta_k}$, and let $\widehat{B}_k$ be the batch obtained by greedy maximization of the lagged objective $\widetilde{H}_{\theta_{k-\ell_k}}$. Under Assumption A.3,*

$$\widetilde{H}_{\theta_k}(\widehat{B}_k) \ge (1 - 1/e)\,\widetilde{H}_{\theta_k}(B_k^\star) - 2b(2 + 2\alpha)\ell_k\epsilon. \tag{66}$$

*Proof.* By the standard greedy guarantee for monotone submodular maximization (Corollary 5.12) on the lagged objective:

$$\widetilde{H}_{\theta_{k-\ell_k}}(\widehat{B}_k) \ge (1 - 1/e)\,\widetilde{H}_{\theta_{k-\ell_k}}(B_k^\star). \tag{67}$$

Applying Lemma A.5 to relate the lagged objective back to the current parameters $\theta_k$:

$$\begin{aligned}
\widetilde{H}_{\theta_k}(\widehat{B}_k) &\ge \widetilde{H}_{\theta_{k-\ell_k}}(\widehat{B}_k) - b(2 + 2\alpha)\ell_k\epsilon \\
&\ge (1 - 1/e)\,\widetilde{H}_{\theta_{k-\ell_k}}(B_k^\star) - b(2 + 2\alpha)\ell_k\epsilon \\
&\ge (1 - 1/e)\left(\widetilde{H}_{\theta_k}(B_k^\star) - b(2 + 2\alpha)\ell_k\epsilon\right) - b(2 + 2\alpha)\ell_k\epsilon \\
&= (1 - 1/e)\,\widetilde{H}_{\theta_k}(B_k^\star) - \underbrace{\left((1 - 1/e) + 1\right)}_{<2} b(2 + 2\alpha)\ell_k\epsilon \\
&\ge (1 - 1/e)\,\widetilde{H}_{\theta_k}(B_k^\star) - 2b(2 + 2\alpha)\ell_k\epsilon. 
\end{aligned} \tag{68}$$

$\square$

*Remark* A.7. We make the following two remarks:

- For **HOBIT**, where embeddings are fixed to the state at the end of the previous epoch, the lag grows linearly with the step count within the current epoch, bounded by $\ell_k \le t$. Here, $t = \frac{|D_{\text{trn}}|}{b}$ denotes the number of gradient steps in an epoch of training the dual encoder.
- For **HOBIT-C**, where embeddings are retrieved from a rolling cache populated during the previous epoch, an embedding used at step $k$ may originate from any point in the previous epoch. In the worst case, the lag is bounded by $\ell_k \le 2t$.

### A.7. Evaluation Metrics

We evaluate retrieval performance using Mean Reciprocal Rank@$k$ (MRR@$k$), normalized Discounted Cumulative Gain@$k$ (nDCG@$k$), Precision@$k$ (P@$k$), and Recall@$k$ (R@$k$), following standard practice in dense retrieval (Monath et al., 2023; Yang et al., 2024). Metrics are reported for $k \in \{10, 100\}$.

Let $\mathcal{Q}^{\text{tst}}$ denote the test query set with $n = |\mathcal{Q}^{\text{tst}}|$ queries. For a query $q \in \mathcal{Q}^{\text{tst}}$, let $\mathcal{R}_q^k$ be the ranked list of the top-$k$ retrieved documents from the corpus $D$, and let $\mathcal{G}_q$ denote the set of relevant documents for $q$.

**Mean Reciprocal Rank.**

$$\text{MRR@}k = \frac{1}{n} \sum_{q \in \mathcal{Q}^{\text{tst}}} \frac{1}{\min \left\{ \text{rank}(d) \mid d \in \mathcal{R}_q^k \cap \mathcal{G}_q \right\}} \tag{69}$$

where the reciprocal rank is defined to be zero if no relevant document appears within the top-$k$ retrieved results.

**Precision and Recall.**

$$\text{P@}k = \frac{1}{n} \sum_{q \in \mathcal{Q}^{\text{tst}}} \frac{\left| \mathcal{R}_q^k \cap \mathcal{G}_q \right|}{k}, \tag{70}$$

$$\text{R@}k = \frac{1}{n} \sum_{q \in \mathcal{Q}^{\text{tst}}} \frac{\left| \mathcal{R}_q^k \cap \mathcal{G}_q \right|}{\left| \mathcal{G}_q \right|}. \tag{71}$$

**Normalized Discounted Cumulative Gain.** For a query $q$, DCG@$k$ is defined as

$$\text{DCG}_q\text{@}k = \sum_{i=1}^{k} \frac{\mathbb{I}[d_i \in \mathcal{G}_q]}{\log_2(i+1)}, \tag{72}$$

where $d_i$ is the document ranked at position $i$. The ideal DCG, $\text{IDCG}_q\text{@}k$, is computed by ranking all relevant documents for $q$ at the top. The normalized metric is then

$$\text{nDCG@}k = \frac{1}{n} \sum_{q \in \mathcal{Q}^{\text{tst}}} \frac{\text{DCG}_q\text{@}k}{\text{IDCG}_q\text{@}k}. \tag{73}$$

### A.8. Large-Scale NOMIC and MTEB Evaluation

We evaluate scalability on the NOMIC training mixture, which contains 1.6M query–document pairs from 10 sources. We fine-tune `nomic-embed-text-v1-unsupervised` (Nussbaum et al., 2025) for one epoch, changing only the batching strategy. This isolates the effect of batch construction from model architecture, training duration, and optimizer choices.

**Source-aware batch construction.** The NOMIC mixture combines heterogeneous data sources, so examples from different sources should not automatically be treated as valid negatives. We therefore run `HOBIT` independently within each source. Each mini-batch samples four sources and draws 64 examples per source, yielding a batch size of 256 while preserving source-local negative semantics.

**Evaluation protocol.** After fine-tuning, we evaluate retrieval performance on MTEB (Muennighoff et al., 2023) using nDCG@10. Table 8 reports per-task results for random batching and `HOBIT`, along with the absolute improvement.

*Table 8.* Large-scale evaluation on MTEB retrieval tasks after fine-tuning on the 1.6M-pair NOMIC training mixture. Scores are nDCG@10; $\Delta$ reports `HOBIT` − Random.

| Task | Random | HOBIT | $\Delta$ | Task | Random | HOBIT | $\Delta$ |
|---|---|---|---|---|---|---|---|
| ArguAna | 54.20 | 52.21 | -1.99 | HotpotQA | 54.47 | 59.23 | +4.76 |
| CQADupStack | 39.07 | 41.44 | +2.37 | MSMARCO | 31.04 | 34.72 | +3.68 |
| ClimateFEVER | 17.83 | 23.80 | +5.97 | NFCorpus | 33.53 | 34.87 | +1.34 |
| DBPedia | 36.41 | 37.49 | +1.08 | NQ | 40.10 | 46.99 | +6.89 |
| FEVER | 54.84 | 61.04 | +6.20 | Quora | 88.27 | 88.73 | +0.46 |
| FiQA2018 | 34.45 | 36.28 | +1.83 | SCIDOCS | 19.73 | 20.17 | +0.44 |
| SciFact | 70.73 | 70.55 | -0.18 | TRECCOVID | 56.06 | 63.23 | +7.17 |
| Touche2020 | 15.43 | 18.70 | +3.27 | Average | 43.08 | 45.96 | +2.88 |

Table 8 shows that HOBIT improves 13 of 15 retrieval tasks, with an average gain of +2.88 nDCG@10 over random batching. The gains are statistically significant under a paired $t$-test across tasks ($p = 0.001$). The only exceptions are ArguAna and SciFact, where random batching is slightly better.

### A.9. Validating Lagged Embeddings and Warmup

Assumption A.3 is used only to analyze HOBIT-C; the batching algorithm itself does not require the assumption. To empirically check whether early training instability affects HOBIT, we compare the default setting, which applies HOBIT from the first epoch, with a warmup variant that uses random batches for the first $R_w = 4$ epochs and then switches to HOBIT.

*Table 9.* Warmup ablation for HOBIT with roberta-base. $R_w = 0$ is the default; $R_w = 4$ uses random batches for the first four epochs.

| Dataset | Metric | $R_w = 0$ | $R_w = 4$ |
|---|---|---|---|
| NQ | R@10 | 52.4 | 52.0 |
| MSMARCO | MRR@10 | 27.4 | 26.8 |
| TriviaQA | R@10 | 52.5 | 51.4 |

Table 9 shows that warmup does not improve retrieval performance. This suggests that, starting from pretrained encoders, the embedding space is stable enough for HOBIT to be useful from epoch 1. For HOBIT-C, Appendix A.6 provides the formal lagged-embedding analysis: cached embeddings from the previous epoch introduce bounded staleness when step-wise embedding drift is small.

### A.10. Batch-Size Sensitivity

We evaluate whether HOBIT remains useful as the batch size increases. Larger random batches are more likely to contain informative negatives by chance, so this ablation tests whether explicit batch construction continues to provide value beyond this effect.

*Table 10.* Batch-size sensitivity on NQ with roberta-base. Scores are R@10.

| Batch size | Random | HOBIT | Gain |
|---|---|---|---|
| 64 | 44.5 | 51.3 | +6.8 |
| 128 | 45.1 | 52.4 | +7.3 |
| 256 | 46.9 | 54.1 | +7.2 |
| 512 | 47.3 | 54.8 | +7.5 |

As shown in Table 10, both Random and HOBIT improve with larger batches, but HOBIT maintains a consistent +6.8 to +7.5 R@10 advantage. This indicates that explicit hardness-aware batch construction remains beneficial even when random batching has more opportunities to sample hard negatives.

### A.11. Seed Selection Ablation

We also vary how the initial seed set is chosen before greedy batch completion. This ablation concerns the seed initialization strategy, which is distinct from the seed set size $s$ studied in Section 6.8. We compare the default random strategy with two coreset-style alternatives: K-means, which chooses seeds from distinct query-embedding clusters, and facility location, which chooses representative seeds under a coverage objective.

*Table 11.* Seed initialization ablation on NQ with roberta-base.

| Seed strategy | R@10 | Epochs |
|---|---|---|
| Random | 52.36 | 20 |
| K-means | 52.44 | 18 |
| Facility location | 52.43 | 16 |

Table 11 shows that all seed strategies perform within 0.07 R@10. Although coreset-style seed selection can reduce the number of epochs, it adds clustering or facility-location overhead. We therefore keep random seed selection as the default because it is simple, inexpensive, and essentially matches the retrieval quality of the more complex alternatives.

### A.12. Effect of Missing Labels on TriviaQA

We complement the NQ sparse-label analysis in Section 6.6 with the same experiment on TriviaQA. As in the main text, we restrict the number of positives available to the batch sampler to $k \in \{2, 4, 8, 16, 32, 64\}$ while keeping the training loss and evaluation protocol unchanged.

Figure 5 shows that HOBIT is robust to missing relevance labels on TriviaQA as well. Performance improves as more positives are available to the sampler, and for moderate values of $k$, HOBIT remains above the strongest non-HOBIT batch sampler on both Recall@10 and Recall@100.

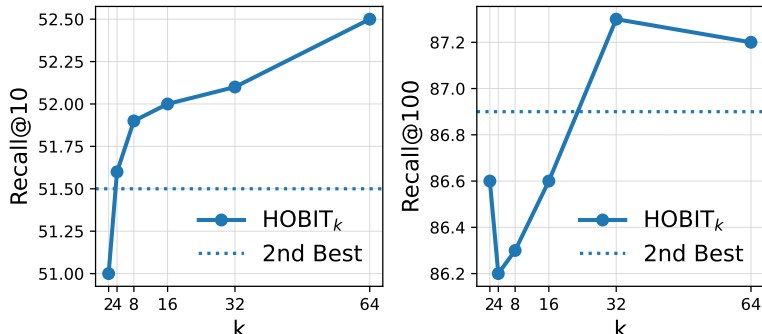

*Figure 5.* Effect of missing relevance labels on HOBIT$_k$ for TriviaQA. Recall@10 (left) and Recall@100 (right) are shown, with dotted lines marking the strongest non-HOBIT batch sampler.

### A.13. Extended Evaluation

In this appendix, we report extended retrieval metrics for all batch construction methods across datasets and encoder architectures. While the main paper focuses on primary metrics (MRR@10 for MSMARCO and Recall@10/100 for NQ and TriviaQA), these tables provide a more comprehensive view of retrieval behaviour across ranking depths.

For each dataset, we report nDCG@k, Recall@k, and MRR@k for $k \in \{1, 5, 10, 20, 100\}$. All results follow the same training and evaluation protocol described in Section 6.1. For clarity, we group results by encoder and highlight HOBIT in blue.

*Table 12.* Extended retrieval metrics on MSMARCO Passage. We report nDCG@k, Recall@k, and MRR@k for $k \in \{1, 5, 10, 20, 100\}$ across batch construction methods and encoders. Results are grouped by encoder, with HOBIT highlighted.

| Method | nDCG@1 | nDCG@5 | nDCG@10 | nDCG@20 | nDCG@100 | R@1 | R@5 | R@10 | R@20 | R@100 | MRR@1 | MRR@5 | MRR@10 | MRR@20 | MRR@100 |
|---|---|---|---|---|---|---|---|---|---|---|---|---|---|---|---|
| | | | | | | roberta-base | | | | | | | | | |
| Random | 14.7 | 27.2 | 30.8 | 33.7 | 37.4 | 14.3 | 38.8 | 49.9 | 61.2 | 80.4 | 14.7 | 23.7 | 24.4 | 25.0 | 25.7 |
| NGAME | 15.8 | 28.8 | 32.8 | 35.6 | 39.2 | 15.3 | 40.8 | 52.8 | 63.8 | 83.1 | 15.8 | 25.2 | 26.8 | 27.6 | 28.1 |
| BatchSampler | 16.2 | 29.2 | 33.0 | 35.9 | 39.5 | 15.7 | 41.4 | 52.9 | 64.1 | 83.2 | 16.2 | 23.9 | 24.2 | 25.2 | 25.4 |
| B3 | 13.9 | 25.2 | 28.6 | 31.5 | 35.4 | 13.5 | 35.8 | 46.3 | 57.4 | 78.1 | 13.9 | 22.0 | 23.4 | 24.2 | 24.7 |
| HOBIT | **16.3** | **29.4** | **33.4** | **36.2** | **39.6** | **15.7** | **41.6** | **53.7** | **64.6** | 82.8 | **16.3** | **25.8** | **27.4** | **28.2** | **28.6** |
| | | | | | | e5-large-unsupervised | | | | | | | | | |
| Random | 15.5 | 29.6 | 33.7 | 36.6 | 40.2 | 15.1 | 42.8 | 55.4 | 66.7 | 85.3 | 15.5 | 25.5 | 27.2 | 28.0 | 28.5 |
| NGAME | 16.4 | 30.9 | 35.3 | 38.3 | 41.7 | 16.0 | 44.2 | 57.4 | 69.1 | 87.2 | 16.4 | 26.8 | 28.6 | 29.5 | 29.9 |
| BatchSampler | 15.0 | 29.3 | 33.5 | 36.3 | 39.9 | 14.5 | 42.7 | 55.5 | 66.2 | 85.3 | 15.0 | 25.2 | 27.0 | 27.7 | 28.2 |
| B3 | 15.3 | 29.4 | 33.4 | 36.3 | 39.9 | 14.9 | 42.5 | 54.6 | 65.9 | 84.8 | 15.3 | 25.4 | 27.0 | 27.8 | 28.3 |
| HOBIT | **18.1** | **32.3** | **36.6** | **39.7** | **43.0** | **17.6** | **45.5** | **58.6** | **70.4** | **87.8** | **18.1** | **28.3** | **30.1** | **30.9** | **31.3** |

*Table 13.* Extended retrieval metrics on Natural Questions (NQ). We report nDCG@k, Recall@k, and MRR@k for $k \in \{1, 5, 10, 20, 100\}$ across batch construction methods and encoders. Results are grouped by encoder, with HOBIT highlighted.

| Method | nDCG@1 | nDCG@5 | nDCG@10 | nDCG@20 | nDCG@100 | R@1 | R@5 | R@10 | R@20 | R@100 | MRR@1 | MRR@5 | MRR@10 | MRR@20 | MRR@100 |
|---|---|---|---|---|---|---|---|---|---|---|---|---|---|---|---|
| | | | | | | roberta-base | | | | | | | | | |
| Random | 13.2 | 23.9 | 27.7 | 30.8 | 34.9 | 11.6 | 33.9 | 45.1 | 56.9 | 77.8 | 13.2 | 21.7 | 23.2 | 24.0 | 24.6 |
| NGAME | 15.3 | 25.6 | 29.8 | 32.8 | 36.7 | 13.5 | 35.5 | 47.9 | 59.2 | 78.9 | 15.3 | 23.7 | 25.4 | 26.2 | 26.7 |
| BatchSampler | 15.7 | 26.5 | 30.7 | 33.9 | 37.9 | 13.9 | 36.6 | 49.1 | 61.3 | 81.5 | 15.7 | 24.4 | 26.1 | 26.9 | 27.4 |
| B3 | 13.5 | 23.0 | 27.2 | 30.2 | 34.2 | 11.9 | 32.2 | 44.5 | 55.8 | 76.1 | 13.5 | 21.2 | 22.9 | 23.7 | 24.2 |
| HOBIT | **16.6** | **28.3** | **32.7** | **35.8** | **39.6** | **14.9** | **39.3** | **52.4** | **63.7** | **83.2** | **16.6** | **25.9** | **27.6** | **28.4** | **28.9** |
| | | | | | | e5-large-unsupervised | | | | | | | | | |
| Random | 18.0 | 31.0 | 35.3 | 38.9 | 42.7 | 15.7 | 43.3 | 56.1 | 69.6 | 88.5 | 18.0 | 28.4 | 30.2 | 31.1 | 31.5 |
| NGAME | 20.4 | 34.4 | 39.1 | 42.3 | 45.7 | 17.9 | 47.6 | 61.3 | 73.5 | 90.5 | 20.4 | 31.6 | 33.4 | 34.2 | 34.6 |
| BatchSampler | 17.6 | 31.0 | 35.6 | 39.0 | 42.7 | 15.6 | 43.7 | 57.4 | 70.0 | 88.3 | 17.6 | 28.1 | 30.0 | 30.9 | 31.3 |
| B3 | 18.8 | 33.0 | 37.9 | 41.2 | 44.7 | 16.6 | 46.1 | 60.4 | 72.5 | 90.0 | 18.8 | 30.1 | 32.1 | 32.9 | 33.3 |
| HOBIT | **22.8** | **36.5** | **41.1** | **44.2** | **47.3** | **20.0** | **49.5** | **62.9** | **74.5** | 89.9 | **22.8** | **34.0** | **35.7** | **36.5** | **36.8** |

*Table 14.* Extended retrieval metrics on TriviaQA. We report nDCG@k, Recall@k, and MRR@k for $k \in \{1, 5, 10, 20, 100\}$ across batch construction methods and encoders. Results are grouped by encoder, with HOBIT highlighted.

| Method | nDCG@1 | nDCG@5 | nDCG@10 | nDCG@20 | nDCG@100 | R@1 | R@5 | R@10 | R@20 | R@100 | MRR@1 | MRR@5 | MRR@10 | MRR@20 | MRR@100 |
|---|---|---|---|---|---|---|---|---|---|---|---|---|---|---|---|
| | | | | | | roberta-base | | | | | | | | | |
| Random | 63.2 | 59.9 | 59.7 | 61.2 | 67.8 | 12.0 | 35.3 | 48.7 | 62.2 | 85.1 | 63.2 | 71.2 | 71.8 | 72.1 | 72.2 |
| NGAME | 66.6 | 63.1 | 62.9 | 64.6 | 70.7 | 12.9 | 37.5 | 51.5 | 65.5 | 86.9 | 66.6 | 74.1 | 74.7 | 75.0 | 75.1 |
| BatchSampler | 63.4 | 59.6 | 59.4 | 61.0 | 67.6 | 12.1 | 35.0 | 48.4 | 61.9 | 85.0 | 63.4 | 71.0 | 71.8 | 72.0 | 72.1 |
| B3 | 62.1 | 58.4 | 58.0 | 59.3 | 65.8 | 11.7 | 34.0 | 46.9 | 60.0 | 82.9 | 62.1 | 69.9 | 70.6 | 70.9 | 71.0 |
| HOBIT | **67.4** | **64.2** | **64.0** | **65.5** | **71.4** | **13.2** | **38.5** | **52.5** | **66.2** | **87.2** | **67.4** | **75.0** | **75.5** | **75.7** | **75.8** |
| | | | | | | e5-large-unsupervised | | | | | | | | | |
| Random | 67.4 | 64.7 | 64.5 | 66.1 | 72.4 | 13.0 | 38.6 | 53.2 | 67.1 | 89.2 | 67.4 | 74.8 | 75.5 | 75.7 | 75.8 |
| NGAME | 71.0 | 68.0 | 68.0 | 69.8 | 75.5 | 14.3 | 41.6 | 56.0 | 70.8 | 90.1 | 71.0 | 77.9 | 78.5 | 78.7 | 78.7 |
| BatchSampler | 67.7 | 64.2 | 64.1 | 65.7 | 72.0 | 13.0 | 38.6 | 52.9 | 66.7 | 88.8 | 67.7 | 74.7 | 75.3 | 75.5 | 75.6 |
| B3 | 67.4 | 64.0 | 63.7 | 65.1 | 71.4 | 13.1 | 38.5 | 52.5 | 65.9 | 87.9 | 67.4 | 74.6 | 75.2 | 75.5 | 75.6 |
| HOBIT | **72.8** | **69.6** | **69.7** | **71.3** | **76.7** | **14.7** | **42.5** | **57.7** | **72.0** | **91.2** | **72.8** | **79.5** | **80.0** | **80.2** | **80.2** |

