# OpenReview forum: "HOBIT: Hardness Optimized Batch Sampling for InfoNCE Training"
_ICML.cc/2026/Conference — ICML 2026 spotlight_

### Official Review · Reviewer_3DbE · 2026-03-08

**Soundness:** 4
**Presentation:** 4
**Significance:** 4
**Originality:** 4
**Overall Recommendation:** 5
**Confidence:** 3

**Summary:**

The authors provide a new batch construction method, called HOBIT, that improves in-batch negatives by making sure that each query in the batch is exposed to hard, but non-contradictory negative examples. They do so by framing the batch construction problem as a monotone submodular optimization problem, for which a simple greedy algorithm achieves a $1-1/e$ approximation guarantee. The authors complement their proposal with a suite of experiments, which showcase that HOBIT outperforms existing batching methods.

**Compliance With Llm Reviewing Policy:**

Affirmed.

**Key Questions For Authors:**

1) It seems to me that the $\tau$ in the InfoNCE loss need not be the same as the $\tau$ when used to replace the hard max with the smooth Log-Sum-Exp function. However, in your algorithm, I believe you set these two $\tau$'s to be the same. Is there any reason we should set these to be the same (i.e. what is the intuition beyond your design choice to set these to be the same)? What happens if these two are set differently? How should a practitioner set these two? Should they always be set to be equal?

2) I don't see where $\alpha$ is used in the ConstructBatch algorithm? Should it be $\alpha = 200$ not $k = 200$?

**Limitations:**

yes

**Strengths And Weaknesses:**

Strengths:
- This paper is extremely well-written and very easy to read; as a non-expert, I felt that the authors did a great job at motivating the problem and providing step-by-step intuition behind their proposed method. In addition, I felt that the authors were very thorough with their explanations, opting to provide derivations whenever possible rather than vague explanations. As such, I felt that the proposed method was properly theoretically justified, as opposed to a sequence of arbitrary algorithmic choices.
- The proposed method is simple and intuitive, which I take as a big plus
- Experimental results seem promising

Weaknesses: I don't have any major weaknesses that come to mind. That said, I do think it would be good if the authors could provide more details about the hard-negative mining approaches in the main text.

---

> ### Author Rebuttal · Authors · 2026-03-31
>
> We thank the reviewer for a positive review and appreciation of our work. Please see below for our responses and we are happy to provide further clarifications.
>
> # W1: More details about the hard-negative mining approaches
>
> We thank the reviewer for this suggestion. Below, we provide a brief description of the hard-negative baselines. We will include a more detailed discussion in Section 6 (Experiments) of the revised version.
>
> - **ANCE:** ANCE constructs an approximate nearest neighbor search (ANNS) index (e.g., FAISS) over the entire document corpus. For each $(q, d^+)$ pair, it retrieves a hard negative by querying the index: $d^- = \arg\max_{d \in D \setminus \{d^+\}} \; q^\top d$, i.e., the most similar (non-positive) document to the query. For MSMARCO, as discussed in our paper, this index is built over millions of document embeddings (approximately 8M). Since rebuilding the index at every epoch is computationally expensive, ANCE refreshes the index periodically, once every $R$ epochs. In our implementation, we set $R = 1$ for NQ and TQA, and $R = 2$ for MSMARCO.
>
> - **TriSampler:** TriSampler applies a quasi-triangular principle to select negatives that are informative yet unlikely to be false negatives. It operates in two stages. First, for each query $q$, it retrieves the top-$K$ most similar documents from the corpus. Then, it applies two sampling distributions sequentially: (1) $p(d^-) \propto \exp\!\left(-\tfrac{1}{4}(s(q, d^-) - s(q, d^+))^2\right)$, which ensures the negative has similar query-relevance to the positive (avoiding too-hard or too-easy negatives); and (2) $p(d^-) \propto \text{ReLU}(s(d^+, d^-) - s(q, d^-))$, which prioritizes negatives closer to the positive document than to the query, constraining the selection within a triangular region in embedding space.
>
> # Q1:  $\tau$ in InfoNCE vs $\tau_h$ in HOBIT
>
> The two temperatures, $\tau$ (InfoNCE) and $\tau_h$ (HOBIT) need **not** necessarily be tied. However, in practice, setting $\tau_h = \tau = 0.05$ works well and requires no additional tuning.
>
> This choice is both empirically and intuitively justified.
>
> - $\tau = 0.05$ for InfoNCE is used predominantly in prior work and several popular codebases. Further, our experiments in Fig. 4 confirm that $\tau_h=0.05$  yields the best performance.
>
> - The parameter $\tau_h$ controls how hardness is measured during batch construction: small $\tau_h$ emphasizes the single hardest negative (hard-max behavior), while larger $\tau_h$ captures average hardness across negatives.  The InfoNCE loss also exhibits a similar dependence on $\tau$: smaller $\tau$ concentrates the loss on the most dominant negative (provided it exceeds the query–positive score), whereas larger $\tau$ distributes weight across multiple negatives.
> We wanted to align batch construction with how the training objective perceives hardness. To this end, we set $\tau_h = \tau$, ensuring both operate on the same hardness scale thereby eliminating the need for additional hyperparameter tuning.
>
>
>
> # Q2:  $\alpha$ in `ConstructBatch` Alg
>
> In Algorithm 2, $\alpha$ does not appear explicitly, as it is incorporated into the hardness score computation in Line 9 via Equation 16:
> $w_{ij} = s(q_i, d_j^+) - \alpha \cdot s(d_i^+, d_j^+)$.
>
> The parameter $k = 200$ in Line 6 is unrelated to $\alpha$; it denotes the *top-$k$ candidate pool size*, i.e., the number of nearest neighbors considered for each seed query. We will clarify this distinction in the revised manuscript.
>
> In our experiments, we use a default of $\alpha = 1.0$. As shown in our ablation study (Table 4), this setting achieves the best performance (Recall@10 = 52.37). Additionally, following Reviewer `2gCh`'s suggestion, we evaluated an $\alpha$-annealing schedule (see response to W3). We observed that such an annealing scheme also yields comparable performance, making HOBIT robust to several different choices for $\alpha$. We will include these clarifications in the revision.
>
> ---
>
> > In response to Reviewer `qnMz` Q4, we conducted a large-scale experiment (1.6M pairs), further strengthening HOBIT’s effectiveness. We request the reviewer to take a look.

---

> > ### Author Rebuttal · Reviewer_3DbE · 2026-04-02
> >
> > Thanks for the response! I will keep my positive score.

---

> > > ### Author Response · Authors · 2026-04-04
> > >
> > > We thank the reviewer for the positive review. We plan the following revisions based on your feedback:
> > >
> > > 1. **Hard-negative mining details (W1):** We will include a dedicated discussion of ANCE and TriSampler in Section 6 of the main text, covering their mechanisms, computational trade-offs, and how they complement batch construction methods like HOBIT.
> > > 2. **$\tau_h$ vs $\tau$ clarification (Q1):** We will clarify in the revised manuscript that $\tau_h$ and $\tau$ need not be tied, while noting that setting $\tau_h = \tau$ is empirically well-motivated and eliminates an extra hyperparameter. The limiting behavior: as $\tau_h \to 0$, the objective approaches the hard-max (optimizing for the single hardest negative per query), and as $\tau_h \to \infty$, it optimizes average batch hardness.
> > > 3. **$\alpha$ in Algorithm 2 (Q2):** We will clarify the distinction between $\alpha$ (hardness score parameter) and $k$ (top-$k$ candidate pool size) in the algorithm description.
> > >
> > > Together with the new large-scale results (Reviewer `qnMz` Q4), we believe these revisions further strengthen the paper. We would be grateful if the reviewer considers these additions in the final assessment.

---

### Official Review · Reviewer_2gCh · 2026-03-11

**Soundness:** 3
**Presentation:** 4
**Significance:** 3
**Originality:** 3
**Overall Recommendation:** 4
**Confidence:** 3

**Summary:**

The paper introduces a novel mini-batch construction method to improve contrasive training with InfoNCE loss. Random batching often lacks informative negative samples and that explicit hard-negative mining can be computationally expensive. To overcome these drawbacks, the authors propose a method that dynamically reorders training examples to create optimal mini-batches. Since directly maximizing the batch hardness is shown to be NP-hard, the authors introduce a smooth Log-Sum-Exp surrogate loss. They prove this surrogate is monotone and submodular, which allows for a greedy approximation algorithm with standard theoretical guarantees. Furthermore, to mitigate the computational cost of continuous embedding updates, they introduce HOBIT-C, a cached variant. Empirical evaluations on MSMARCO, Natural
Questions, and TriviaQA demonstrate that HOBIT outperforms existing batching methods, complements explicit hard-negative mining techniques, and maintains robustness under sparse relevance labels.

**Compliance With Llm Reviewing Policy:**

Affirmed.

**Final Justification:**

The authors have addressed all my concerns. I maintain my original score.

**Key Questions For Authors:**

1. Table 5 shows the efficiency of HOBIT-C relative to HOBIT. Could you provide more information (e.g., steps per second or total epoch time) comparing Random Batching, NGAME, vanilla HOBIT, and HOBIT-C? Clarifying this overhead is essential for assessing practical utility.

2. In numerical experiments, the default choice is $\alpha=1.0$. Since the embedding space is highly uninformative in the first few epochs, did you experiment with annealing $\alpha$ (e.g., starting near 0 and scaling up to 1) to prevent the non-contradiction penalty from introducing noise early in training?

3. You use a batch size of 128 following prior works. Will HOBIT's advantage over Random Batching change with batch size? As batch size increases, the chance of randomly sampling a hard negative also increases. It might be worthwhile to see if the HOBIT's advantage change with batch size.

**Limitations:**

Yes

**Strengths And Weaknesses:**

Strengths:

The manuscript has a nice presentation of problem formulation, starting from introducing InfoNCE objective to introducing two complementary conditions for the in-batch negatives, and then motivate the HOBIT objective. The manuscript has conducted extensive numerical comparison with existing baselines across multiple datasets and encoder architectures (RoBERTa and E5). Dual-encoders are the backbone of modern large-scale retrieval systems. Improving their training efficiency and final performance offers substantial practical utility to the community. While batch sampling for contrastive learning is not entirely new, the proposed method provides an efficient mini-batch construction method that can maintain in-batch negative quality along the iterations.

Weakness:

Replacing a hard max function with a smooth Log-Sum-Exp function is not new in the machine learning literature. Such a surrogate has been used in other ML problems.

The theoretical guarantees for HOBIT-C rely heavily on Assumption A.3 (bounded step-wise drift). In the early epochs of contrastive fine-tuning, embedding spaces may undergo drastic shifts due to high gradient variance. It implies that this assumption could be violated in early epochs. Additionally, the hardness score relies on a static balancing parameter $\alpha$. A fixed $\alpha$ might not be optimal across the entire training trajectory.

---

> ### Author Rebuttal · Authors · 2026-03-31
>
> Thank you for the feedback and insightful questions.
>
> # W1: Log-Sum-Exp
> We agree that Log-Sum-Exp (LSE) as a max surrogate is standard in optimization [1, §3.1.5]. Our contribution, however, is not the use of LSE surrogate itself, but its *specific formulation, analysis, and application* in the context of InfoNCE-based contrastive learning. Concretely:
> - We show that batch construction without LSE is NP-hard.
> - We linked LSE relaxation to InfoNCE loss, tying gradient magnitude to in-batch negative hardness.
> - This exposes a limitation of existing hard negative objectives were they ignore the non-contradictory penalty (i.e., do not penalize near-positive negatives).
> - We prove the objective is monotone submodular, enabling a greedy $(1-1/e)$ approximation.
>
> To our knowledge, this perspective is novel and unexplored in prior work on negative mining and batch sampling.
>
> # W2: Assumption A.3
> This assumption is used only for analyzing HOBIT-C; the algorithm itself does not depend on it. All experiments start from pretrained checkpoints (RoBERTa-base, e5-large-unsupervised, nomic-text-v1-unsupervised), so embeddings are well-formed even in early epochs, making drift less pronounced than with random initialization.
>
> To validate this further, we introduce a warmup parameter $R_w$: batches are random for the first $R_w$ epochs, then HOBIT is applied. Comparing $R_w=0$ (default) vs $R_w=4$ (10% epochs) on RoBERTa-base:
>
> |Dataset|HOBIT|HOBIT ($R_w = 4$)|
> |-|-|-|
> |NQ (R@10)|52.4|52.0|
> |MSMARCO (MRR@10)|27.4|26.8|
> |TriviaQA (R@10)|52.5|51.4|
>
> $R_w=0$ consistently outperforms $R_w=4$, suggesting optimizing batches from epoch 1 is preferable to random warmup, supporting Assumption A.3 in practice.
>
> # W3: fixed $\alpha$
> We study sensitivity to fixed $\alpha$ on NQ with RoBERTa-base:
>
> |$\alpha$|R@10|
> |-|-|
> |0.1|50.1|
> |0.5|50.1|
> |0.9|51.8|
> |1.0 (default)|52.4|
>
> HOBIT outperforms random batching across all $\alpha$, with performance improving as $\alpha$ increases, indicating stronger penalization of positive–positive similarity is beneficial. We also test linear annealing from $0 \to 1$:
>
> ||R@10|
> |-|-|
> |anneal|52.2|
> |Fixed $\alpha=1.0$|52.4|
>
> Annealing matches but does not exceed the best fixed setting, showing no clear benefit over applying full strength from the start. In summary: (i) larger $\alpha$ helps, (ii) $\alpha=1.0$ throughout is simple and effective.
>
> # Q1: overhead comparison
> We report two components: (i) per-epoch overhead (method-specific preprocessing) and (ii) forward/backward training time (common to all methods).
>
> Per-epoch overhead differs across methods:
> - HOBIT: query and positive embedding generation, batch construction (all-pairs similarity, top-k filtering, and a greedy selection loop).
> - HOBIT-C: reuses embeddings from previous epoch, so overhead is only batch construction.
> - ANCE: query and full-corpus embedding generation, indexing, and ANNS-based retrieval.
> - NGAME: query embedding generation and k-means clustering.
>
> **NQ/RoBERTa-base** (fwd/bwd time: 155s):
> ||Overhead (s)|OH %|R@10|Epochs|Training time (mins)|
> |-|-|-|-|-|-|
> |Random|0|0%|45.1|10|26|
> |NGAME|12|8%|47.9|7|19|
> |HOBIT|35|23%|**52.4**|20|63|
> |HOBIT-C|0.2|0.1%|51.7|14|36|
>
> **MSMARCO/RoBERTa-base** (fwd/bwd time: 460s)
> ||Overhead (s)|OH %|MRR@10|Epoch|Training time (mins)|
> |-|-|-|-|-|-|
> |Random|0|0%|24.4|26|199|
> |NGAME|65|14%|26.8|19|166|
> |HOBIT|130|28%|**27.4**|24|236|
> |HOBIT-C|8|1.7%|27.2|14|109|
> - HOBIT-C adds near-zero overhead, making it 60× and 8× cheaper than NGAME on NQ, MSMARCO respectively.
> - On NQ, HOBIT-C improves R@10 by +6.6 pts offering massive quality gains.
> - On MSMARCO, it gains +2.8 MRR@10 with similar trade-offs.
> - Compared to HOBIT, HOBIT-C is much more efficient, while HOBIT achieves the best absolute performance at higher training cost.
>
> # Q2: annealing $\alpha$
> See W3.
>
> # Q3: large batch sizes
> We agree larger batch sizes help both Random and HOBIT, but for different reasons: Random benefits from higher chances of sampling hard negatives, while HOBIT benefits systematically since a larger candidate pool (with fixed seed size $s$) enables selection of more informative, harder negatives per query. On NQ (RoBERTa-base) with batch sizes $\{64,128,256,512\}$:
>
> |Batch Size|Random (R@10)|HOBIT (R@10)|
> |-|-|-|
> |64|44.5|51.3|
> |128|45.1|52.4|
> |256|46.9|54.1|
> |512|47.3|54.8|
>
> Both improve with batch size, but HOBIT maintains a strong gain (+6.8 to +7.5), showing robust hard-negative selection at scale. Notably, HOBIT at 64 (51.3) outperforms Random at 512 (47.3) by +4.0, highlighting its advantage in memory-constrained settings. Due to GPU limits, we scale up to 512 (1024 causes OOM); at larger sizes, diminishing negative quality may narrow the gap. We will include this in the main paper.
>
> ---
> [1] Boyd S. et al (2004). Convex optimization. Cambridge univ press.
>
> > In response to Reviewer `qnMz` Q4, we conducted a large-scale experiment (1.6M pairs), further strengthening HOBIT’s effectiveness. We request you to take a look.

---

> > ### Author Rebuttal · Reviewer_2gCh · 2026-04-02
> >
> > The rebuttal has adequately addressed all my concerns. I would recommend the authors adding a reference of the Log-Sum-Exp surrogate and incorporating part of their rebuttal to W1 in the revised manuscript.

---

> > > ### Author Response · Authors · 2026-04-04
> > >
> > > We thank the reviewer for the positive assessment and for the thoughtful acknowledgement. We are glad that our rebuttal adequately addressed all concerns. Your suggestions, particularly regarding the $\alpha$ annealing experiment, the batch size sensitivity analysis, and the LSE citation, led to new experiments and improved presentation.
> > >
> > > As suggested, we plan the following revisions:
> > >
> > > 1. **LSE citation (W1):** We will add a proper citation for the Log-Sum-Exp surrogate (Boyd & Vandenberghe, 2004) and incorporate a concise version of our rebuttal to W1 in the revised manuscript, clearly delineating the novelty of our formulation from the standard use of LSE.
> > > 2. **Assumption A.3 validation (W2):** We will include the $R_w$ warmup experiment and the HOBIT vs. HOBIT-C comparison table (across 6 settings) in the appendix, providing concrete empirical support for the bounded drift assumption.
> > > 3. **$\alpha$ annealing (W3):** We will add the annealing experiment results alongside the existing fixed-$\alpha$ ablation in the main paper.
> > > 4. **Overhead analysis (Q1):** We will include the per-component timing tables in the main paper.
> > > 5. **Batch size sensitivity (Q3):** We will include the batch size ablation in the appendix, demonstrating HOBIT's consistent advantage across batch sizes.
> > >
> > > We believe these additions, together with the large-scale NOMIC experiment (Reviewer `qnMz` Q4), substantially strengthen the empirical contribution of our work. We hope the reviewer will consider these improvements when finalizing the assessment.

---

### Official Review · Reviewer_qnMz · 2026-03-12

**Soundness:** 3
**Presentation:** 3
**Significance:** 3
**Originality:** 3
**Overall Recommendation:** 5
**Confidence:** 4

**Summary:**

The paper depicts a batch construction method coined HOBIT that leverages the InfoNCE loss function from the perspective of submodular maximization. Specifically, the authors show that the optimization objective involving the InfoNCE loss function boils down to optimizing a monotone submodular function. To that end, the lazy greedy algorithm is used, which is known to achieve an $O(1-1/e)$ approximation in this setting. Experiments were conducted to demonstrate their efficacy against state-of-the-art methods.

**Compliance With Llm Reviewing Policy:**

Affirmed.

**Final Justification:**

The authors have adequately addressed my questions and concerns, raising my score from 4 to 5.

**Key Questions For Authors:**

1) How does your approach change if you change the initial set $\mathcal{S}$ to be the k-means coreset (sensitivity-based, for example), or other coresets?
2) How does your approach fare when changing the embeddings every R epochs (similar to what was used in GradMatch [1])
3) Can you provide training time, including preprocessing and submodular set construction, against the other approaches to better understand the reach of your gains?
4) Can this approach work well when deployed on very large datasets, for example, $ \geq 1M$ pairs? Or a uniform coreset is needed first to lower the size of the set, followed by HOBIT (or HOBIT-C)?




______________________________________________________________________________________________________
[1] Killamsetty, Krishnateja, et al. "Grad-match: Gradient matching based data subset selection for efficient deep model training." International Conference on Machine Learning. PMLR, 2021.

**Limitations:**

The authors did not adequately discuss the limitations of their approach, one of which is the quadratic time concerning the preprocessing step usually needed for submodular set maximization. Please state the limitations of your approach.

**Strengths And Weaknesses:**

**Strengths**
* The connection between the InfoNCE batch loss and submodularity is intriguing and interesting.
* The hardness score (Definition 4.5) and the geometric intuition behind it, depicted in Figure 2, is informative and well motivated, showcasing the idea that HOBIT aims to leverage.
* The cached version of HOBIT (coined HOBIT-C) provides a cheaper alternative to HOBIT at the expense of a minimal performance quality drop.


**Weaknesses**
* The set $\mathcal{S}$ is initialized with a uniform sample, which is good at early epochs; however, it would have been better to choose a set that would utilize some geometric properties, or even other coreset approaches, as the initial set $\mathcal{S}$.
* Usually, submodular set construction requires quadratic time as a preprocessing step.  How is this reflected when comparing with the other methods?
* The authors did not deeply analyze when their approaches fail or when other approaches might outperform them (also connected to the previous weakness).

---

> ### Author Rebuttal · Authors · 2026-03-31
>
> We thank the reviewer for a comprehensive review of our work.
> # W1: Coreset seed selection
> Thank you for the suggestion. We compare three initializations on NQ (RoBERTa-base):
> - Random
> - K-means: cluster query embeddings at each epoch ($k=128$); pick 8 seeds from distinct clusters
> - Facility location: pick 8 seeds from 128 facilities
>
> |  | R@10 | Epochs |
> |-|-|-|
> | Random | 52.36 | 20 |
> | K-means | 52.44 | 18 |
> | Facility location | 52.43 | 16 |
>
> All methods perform nearly identically (within 0.07). However, coresets converge slightly faster, albeit with increased seed selection overhead. We will include this in the appendix.
> # W2: Submodular cost
> The reviewer is correct: HOBIT, in its current form, incurs an $O(N^2)$ preprocessing cost to compute pairwise similarities. However, for the datasets we consider, this step is efficient with chunked GPU vectorization.
>
> To limit downstream cost, we use top-$k$ filtering (step 6, Alg. 2), shrinking the candidate set and reducing the greedy step to $O(Nk)$. The greedy loop is also vectorized.
>
> As a result, the overall overhead (preprocessing + greedy) remains small compared to training:
> - NQ: 0.2s vs. 155s
> - MSMARCO: 8s vs. 460s
> Importantly, $O(N^2)$ step is only used to get top-$k$ candidates. For larger $N$, it can be replaced with FAISS-based ANN retrieval, avoiding explicit pairwise computation leading to a near-linear scaling $O(Nk)$ for $k << N$. We defer detailed timings to Reviewer `2gCh`(Q1).
> # W3: Failure modes
> We summarize two key limitations:
>
> **1) Higher training time vs. Random**
> HOBIT-C increases end-to-end training time (NQ: 36 vs. 24 min). Although per-epoch costs are similar, the overhead arises from requiring more epochs to converge, as HOBIT constructs consistently harder batches (refer Fig. 1):
> - NQ: 51.7 vs. 45.1 R@10 (+6.6)
>
> HOBIT follows the same trend; further improving performance (52.4 R@10, +0.7 over HOBIT-C) at the cost of additional time (66 min) due to the additional embedding generation step.
>
> **2) Limited gains with strong hard-negative mining**
> When combined with methods such as ANCE, the gains diminish, as strong negatives are already available. We refer the reviewer to Sec. 6.4 for details.
>
> We will include (1) and explicitly highlight (2) as a limitation in the revision.
> # Q1: Initial set via coresets
> Refer W1.
> # Q2: Embedding refresh
> HOBIT-C constructs batches at epoch $i$ using embeddings from epoch $i-1$, so staleness is at most two epochs with no additional embedding generation cost. Unlike GradMatch or ANCE, which trade off larger $R$ for efficiency, HOBIT-C maintains a strict bound of $R=2$ without incurring such trade-offs.
>
> We formalize this in Theorem A.6. Empirically as well, HOBIT-C performs only slightly worse than HOBIT but still outperforms all baselines.
> # Q3: Training time
> Refer `2gCh` (Q1)
> # Q4: Scaling to large datasets
> We evaluate scalability on the **NOMIC dataset**[1] (1.6M $(q,d)$ pairs), comprising a mixture of 10 sources (Reddit, MSMARCO, HotpotQA, etc).
>
> **Setup.** We follow the [NOMIC codebase](https://github.com/nomic-ai/contrastors) and modify only batching. Since valid hard negatives should come from the same source, we adapt HOBIT as follows:
> - Run HOBIT independently within each source (instead of over the full dataset)
> - Sample 4 sources uniformly at random; draw 64 samples from each to form a batch of 256 (NOMIC default)
> - Within each source, HOBIT uses $b=64$, $s=8$
> - All other hyperparameters use defaults ($s=8,\tau=0.05,\alpha=1$)
>
> We fine-tune [nomic-embed-text-v1-unsupervised](https://huggingface.co/nomic-ai/nomic-embed-text-v1-unsupervised) for 1 epoch and evaluate on *MTEB (English v1)*[2], a standard retrieval evaluation benchmark.
> Our implementation is available here: [code for HOBIT applied to NOMIC](https://anonymous.4open.science/r/contrastors_hobit/README.md).
> | Task | HOBIT | Random |
> |-|-|-|
> | *ArguAna* | 52.21 | 54.20 |
> | CQADupstack | 41.44 | 39.07 |
> | ClimateFEVER | 23.80 | 17.83 |
> | DBPedia | 37.49 | 36.41 |
> | FEVER | 61.04 | 54.84 |
> | FiQA2018 | 36.28 | 34.45 |
> | HotpotQA | 59.23 | 54.47 |
> | MSMARCO | 34.72 | 31.04 |
> | NFCorpus | 34.87 | 33.53 |
> | NQ | 46.99 | 40.10 |
> | Quora | 88.73 | 88.27 |
> | SCIDOCS | 20.17 | 19.73 |
> | *SciFact* | 70.55 | 70.73 |
> | TRECCOVID | 63.23 | 56.06 |
> | Touche2020 | 18.70 | 15.43 |
> | Avg (15) | **45.96** | 43.08 |
>
> **Key Findings**
> - HOBIT improves on 13/15 tasks (minor drops on 2 datasets)
> - Avg gain: +2.9 NDCG@10 points
> - A two sided Paired $t$-test yields $p$ value of 0.001 (statistically significant)
>
> We thank the reviewer for this question. These results show that HOBIT scales effectively to large datasets and delivers consistent, statistically significant gains on out-of-distribution benchmarks. This strengthens our work, and we will include these findings in the main paper.
>
> ---
> [1] Nussbaum et al. (2025). "Nomic Embed". TMLR.
>
> [2] Muennighoff et al. (2022). "MTEB: Massive Text Embedding Benchmark". arXiv:2210.07316.

---

> > ### Author Rebuttal · Reviewer_qnMz · 2026-04-02
> >
> > The authors have adequately addressed my questions and concerns. I intend to increase my score to 5.

---

> > > ### Author Response · Authors · 2026-04-04
> > >
> > > We sincerely thank the reviewer for the positive reassessment and for increasing the score. Your feedback led to several valuable experiments, including the coreset-based seed selection ablation, the large-scale NOMIC evaluation, and the detailed overhead analysis, all of which strengthened the paper.
> > >
> > > In the revised manuscript, we plan to incorporate the following:
> > >
> > > 1. **Overhead analysis (W2, Q3):** We will include the per-component timing comparison (Table from Q1 of Reviewer `2gCh`) in the main paper.
> > > 2. **Embedding staleness bound (Q2):** We will crisply highlight the formal staleness guarantee for HOBIT-C ($R = 2$, Theorem A.6) in the main text, alongside the empirical validation across all 6 dataset–model settings.
> > > 3. **Limitations (W3):** We will explicitly include both limitations: (i) the increased wall-clock time due to more convergence epochs, and (ii) the diminished gains when combined with strong explicit hard-negative miners such as ANCE.
> > > 4. **Seed selection ablation (W1):** We will include the coreset seed initialization results in the appendix.
> > > 5. **Large-scale evaluation (Q4):** We will include the NOMIC (1.6M pairs) results and MTEB evaluation in the appendix.
> > >
> > > We are grateful for the constructive engagement and look forward to incorporating these improvements.

---

### Decision · Program_Chairs · 2026-04-30

**Decision:**

Accept (spotlight)

**Comment:**

This paper introduces HOBIT, a principled batch construction method for training dual-encoder models with InfoNCE loss. The core innovation is the formulation of batch selection as a monotone submodular maximization problem, allowing the use of a greedy algorithm with theoretical guarantees to create batches where queries are exposed to "hard but non-contradictory" negatives. The authors propose a cached variant (HOBIT-C) to mitigate computational overhead. The method demonstrates good experimental performance.

In the rebuttal process, all three reviewers engaged substantively with the rebuttal and acknowledged that their concerns were resolved. The final scores are 1x Weak accept and 2x Accept. So I also vote for acceptance.

I recommend that the authors incorporate the new large-scale NOMIC/MTEB results and the detailed overhead timing analysis from the rebuttal into the final camera-ready version to further strengthen the paper's empirical contributions.